# NATURAL- TO FORMAL-LANGUAGE GENERATION USING TENSOR PRODUCT REPRESENTATIONS

## ABSTRACT

Generating formal-language represented by relational tuples, such as Lisp programs or mathematical operations, from natural-language input is a challenging task because it requires explicitly capturing discrete symbolic structural information implicit in the input. Most state-of-the-art neural sequence models do not explicitly capture such structural information, limiting their performance on these tasks. In this paper we propose a new encoder-decoder model based on Tensor Product Representations (TPRs) for Natural- to Formal-language generation, called *TP-N2F*. The encoder of TP-N2F employs TPR 'binding' to encode natural-language symbolic structure in vector space and the decoder uses TPR 'unbinding' to generate, in symbolic space, a sequence of relational tuples, each consisting of a relation (or operation) and a number of arguments. On two benchmarks, TP-N2F considerably outperforms LSTM-based seq2seq models, creating new state-of-the-art results: the MathQA dataset for math problem solving, and the AlgoLisp dataset for program synthesis. Ablation studies show that improvements can be attributed to the use of TPRs in both the encoder and decoder to explicitly capture relational structure to support reasoning.

## 1 INTRODUCTION

When people perform explicit reasoning, they can typically describe the way to the conclusion step by step via relational descriptions. There is ample evidence that relational representations are important for human cognition (e.g., (Goldin-Meadow & Gentner, 2003; Forbus et al., 2017; Crouse et al., 2018; Chen & Forbus, 2018; Chen et al., 2019)). Although a rapidly growing number of researchers use deep learning to solve complex symbolic reasoning and language tasks (a recent review is (Gao et al., 2019)), most existing deep learning models, including sequence models such as LSTMs, do not explicitly capture human-like relational structure information.

In this paper we propose a novel neural architecture, ***TP-N2F***, to solve natural- to formal-language generation tasks (N2F). In the tasks we study, math or programming problems are stated in natural-language, and answers are given as programs, sequences of relational representations, to solve the problem. TP-N2F encodes the natural-language symbolic structure of the problem in an input vector space, maps this to a vector in an intermediate space, and uses that vector to produce a sequence of output vectors that are decoded as relational structures. Both input and output structures are modelled as Tensor Product Representations (TPRs) (Smolensky, 1990). During encoding, NL-input symbolic structures are encoded as vector space embeddings using TPR 'binding' (following Palangi et al. (2018)); during decoding, symbolic constituents are extracted from structure-embedding output vectors using TPR 'unbinding' (following Huang et al. (2018; 2019)).

Our contributions in this work are as follows. (i) We propose a role-level analysis of N2F tasks. (ii) We present a new TP-N2F model which gives a neural-network-level implementation of a model solving the N2F task under the role-level description proposed in (i). To our knowledge, this is the first model to be proposed which combines both the binding and unbinding operations of TPRs to achieve generation tasks through deep learning. (iii) State-of-the-art performance on two recently developed N2F tasks shows that the TP-N2F model has significant structure learning ability on tasks requiring symbolic reasoning through program synthesis.

## 2 BACKGROUND: REVIEW OF TENSOR-PRODUCT REPRESENTATION

The TPR mechanism is a method to create a vector space embedding of complex symbolic structures. The type of a symbol structure is defined by a set of structural positions or roles, such as the left-child-of-root position in a tree, or the second-argument-of-$R$ position of a given relation $R$. In a particular instance of a structural type, each of these roles may be occupied by a particular filler, which can be an atomic symbol or a substructure (e.g., the entire left sub-tree of a binary tree can serve as the filler of the role left-child-of-root). For now, we assume the fillers to be atomic symbols.[1]

The TPR embedding of a symbol structure is the sum of the embeddings of all its constituents, each constituent comprising a role together with its filler. The embedding of a constituent is constructed from the embedding of a role and the embedding of the filler of that role: these are joined together by the TPR 'binding' operation, the tensor (or generalized outer) product $\otimes$.

Formally, suppose a symbolic type is defined by the roles $\{r_i\}$, and suppose that in a particular instance of that type, S, role $r_i$ is bound by filler $f_i$. The TPR embedding of S is the order-2 tensor

$$\mathbf{T} = \sum_i \boldsymbol{f}_i \otimes \boldsymbol{r}_i = \sum_i \boldsymbol{f}_i \boldsymbol{r}_i^\top \tag{1}$$

where $\{\boldsymbol{f}_i\}$ are vector embeddings of the fillers and $\{\boldsymbol{r}_i\}$ are vector embeddings of the roles. In Eq. 1, and below, for notational simplicity we conflate order-2 tensors and matrices.

As a simple example, consider the symbolic type string, and choose roles to be $r_1 = \text{first\_element}$, $r_2 = \text{second\_element}$, etc. Then in the specific string S $=$ cba, the first role $r_1$ is filled by c, and $r_2$ and $r_3$ by b and a, respectively. The TPR for S is $\boldsymbol{c} \otimes \boldsymbol{r}_1 + \boldsymbol{b} \otimes \boldsymbol{r}_2 + \boldsymbol{a} \otimes \boldsymbol{r}_3$, where $\boldsymbol{a}, \boldsymbol{b}, \boldsymbol{c}$ are the vector embeddings of the symbols a, b, c, and $\boldsymbol{r}_i$ is the vector embedding of role $r_i$.

A TPR scheme for embedding a set of symbol structures is defined by a decomposition of those structures into roles bound to fillers, an embedding of each role as a **role vector**, and an embedding of each filler as a **filler vector**. Let the total number of roles and fillers available be $n_\mathrm{R}, n_\mathrm{F}$, respectively. Define the matrix of all possible role vectors to be $\boldsymbol{R} \in \mathbb{R}^{d_\mathrm{R} \times n_\mathrm{R}}$, with column $i$, $[\boldsymbol{R}]_{:i} = \boldsymbol{r}_i \in \mathbb{R}^{d_\mathrm{R}}$, comprising the embedding of $r_i$. Similarly let $\boldsymbol{F} \in \mathbb{R}^{d_\mathrm{F} \times n_\mathrm{F}}$ be the matrix of all possible filler vectors. The TPR $\mathbf{T} \in \mathbb{R}^{d_\mathrm{F} \times d_\mathrm{R}}$. Below, $d_\mathrm{R}, n_\mathrm{R}, d_\mathrm{F}, n_\mathrm{F}$ will be hyper-parameters, while $\boldsymbol{R}, \boldsymbol{F}$ will be learned parameter matrices.

Using summation in Eq.1 to combine the vectors embedding the constituents of a structure risks non-recoverability of those constituents given the embedding $\mathbf{T}$ of the the structure as a whole. The tensor product is chosen as the binding operation in order to enable recovery of the filler of any role in a structure S given its TPR $\mathbf{T}$. This can be done with perfect precision if the embeddings of the roles are linearly independent. In that case the role matrix $\boldsymbol{R}$ has a left inverse $\boldsymbol{U}$: $\boldsymbol{U}\boldsymbol{R} = \boldsymbol{I}$. Now define the **unbinding** (or **dual**) **vector** for role $r_j$, $\boldsymbol{u}_j$, to be the $j^\mathrm{th}$ column of $\boldsymbol{U}^\top$: $U_{:j}^\top$. Then, since $[\boldsymbol{I}]_{ji} = [\boldsymbol{U}\boldsymbol{R}]_{ji} = \boldsymbol{U}_{j:}\boldsymbol{R}_{:i} = [\boldsymbol{U}_{:j}^\top]^\top \boldsymbol{R}_{:i} = \boldsymbol{u}_j^\top \boldsymbol{r}_i = \boldsymbol{r}_i^\top \boldsymbol{u}_j$, we have $\boldsymbol{r}_i^\top \boldsymbol{u}_j = \delta_{ji}$. This means that, to recover the filler of $r_j$ in the structure with TPR $\mathbf{T}$, we can take its tensor inner product (or matrix-vector product) with $\boldsymbol{u}_j$:[2]

$$\mathbf{T}\boldsymbol{u}_j = \left[ \sum_i \boldsymbol{f}_i \boldsymbol{r}_i^\top \right] \boldsymbol{u}_j = \sum_i \boldsymbol{f}_i \delta_{ij} = \boldsymbol{f}_j \tag{2}$$

In the architecture proposed here, we will make use of both TPR binding using the tensor product with role vectors $\boldsymbol{r}_i$ and TPR unbinding using the tensor inner product with unbinding vectors $\boldsymbol{u}_j$. Binding will be used to produce the order-2 tensor $\mathbf{T}_S$ embedding of the NL problem statement. Unbinding will be used to generate output relational tuples from an order-3 tensor $\mathbf{H}$. Because they pertain to different representations (of different orders in fact), the binding and unbinding vectors we will use are not related to one another.

---

[1]When fillers are structures themselves, binding can be used recursively, giving tensors of order higher than 2. In general, binding is done with the tensor product, since conflation with matrix algebra is only possible for order-2 tensors. Our unbinding of relational tuples involves the order-3 TPRs defined in Sec. 3.1.2.

[2]When the role vectors are not linearly independent, this operation performs unbinding approximately, taking $\boldsymbol{U}$ to be the left pseudo-inverse of $\boldsymbol{R}$. Because randomly chosen vectors on the unit sphere in a high-dimensional space are approximately orthogonal, the approximation is often excellent (Anonymous, in prep.).

# 3 TP-N2F MODEL

We propose a general TP-N2F neural network architecture operating over TPRs to solve N2F tasks under a proposed role-level description of those tasks. In this description, natural-language input is represented as a straightforward order-2 role structure, and formal-language relational representations of outputs are represented with a new order-3 recursive role structure proposed here. Figure 1 shows an overview diagram of the TP-N2F model. It depicts the following high-level description.

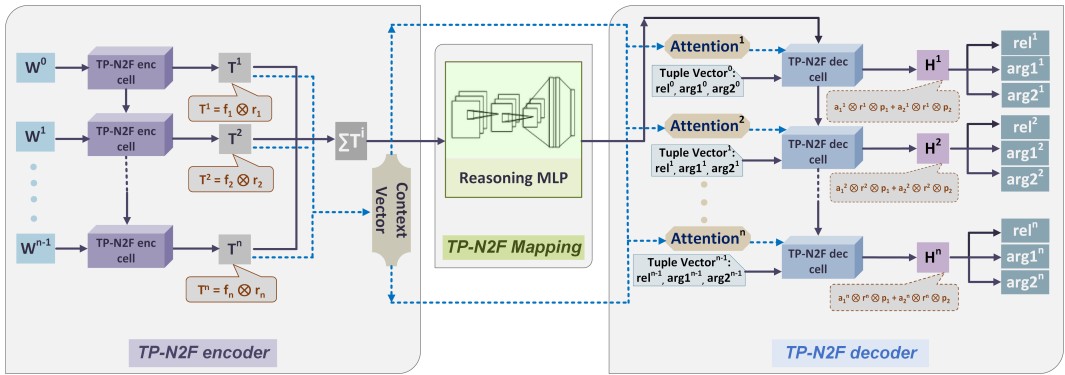

Figure 1: Overview diagram of TP-N2F.

As shown in Figure 1, while the natural-language input is a sequence of words, the output is a sequence of multi-argument relational tuples such as $(R \ A_1 \ A_2)$, a 3-tuple consisting of a binary relation (or operation) $R$ with its two arguments. The "TP-N2F encoder" uses two LSTMs to produce a pair consisting of a filler vector and a role vector, which are bound together with the tensor product. These tensor products, concatenated, comprise the "context" over which attention will operate in the decoder. The sum of the word-level TPRs, flattened to a vector, is treated as a representation of the entire problem statement; it is fed to the "Reasoning MLP", which transforms this encoding of the problem into a vector encoding the solution. This is the initial state of the "TP-N2F decoder" attentional LSTM, which outputs at each time step an order-3 tensor representing a relational tuple. To generate a correct tuple from decoder operations, the model must learn to give the order-3 tensor the form of a TPR for a $(R \ A_1 \ A_2)$ tuple (detailed explanation in Sec. 3.1.2). In the following sections, we first introduce the details of our proposed role-level description for N2F tasks, and then present how our proposed TP-N2F model uses TPR binding and unbinding operations to create a neural network implementation of this description of N2F tasks.

## 3.1 ROLE-LEVEL DESCRIPTION OF N2F TASKS

In this section, we propose a role-level description of N2F tasks, which specifies the filler/role structures of the input natural-language symbolic expressions and the output relational representations.

### 3.1.1 ROLE-LEVEL DESCRIPTION FOR NATURAL-LANGUAGE INPUT

Instead of encoding each token of a sentence with a non-compositional embedding vector looked up in a learned dictionary, we use a learned role-filler decomposition to compose a tensor representation for each token. Given a sentence $S$ with $n$ word tokens $\{w^0, w^1, ..., w^{n-1}\}$, each word token $w^t$ is assigned a learned role vector $\boldsymbol{r}^t$, soft-selected from the learned dictionary $\boldsymbol{R}$, and a learned filler vector $\boldsymbol{f}^t$, soft-selected from the learned dictionary $\boldsymbol{F}$ (Sec. 2). The mechanism closely follows that of Palangi et al. (2018), and we hypothesize similar results: the role and filler approximately encode the grammatical role of the token and its lexical semantics, respectively.[3] Then each word token $w^t$ is represented by the tensor product of the role vector and the filler vector: $\mathbf{T}^t = \boldsymbol{f}^t \otimes \boldsymbol{r}^t$. In addition

---

[3]Although the TPR formalism treats fillers and roles symmetrically, in use, hyperparameters are selected so that the number of available fillers is greater than that of roles. Thus, on average, each role is assigned to more words, encouraging it to take on a more general function, such as a grammatical role.

to the set of all its token embeddings $\{\mathbf{T}^0, \ldots, \mathbf{T}^{n-1}\}$, the sentence $S$ as a whole is assigned a TPR equal to the sum of the TPR embeddings of all its word tokens: $\mathbf{T}_S = \sum_{t=0}^{n-1} \mathbf{T}^t$.

Using TPRs to encode natural language has several advantages. First, natural language TPRs can be interpreted by exploring the distribution of tokens grouped by the role and filler vectors they are assigned by a trained model (as in Palangi et al. (2018)). Second, TPRs avoid the Bag of Word (BoW) confusion (Huang et al., 2018): the BoW encoding of *Jay saw Kay* is the same as the BoW encoding of *Kay saw Jay* but the encodings are different with TPR embedding, because the role filled by a symbol changes with its context.

### 3.1.2 ROLE-LEVEL DESCRIPTION FOR RELATIONAL REPRESENTATIONS

In this section, we propose a novel recursive role-level description for representing symbolic relational tuples. Each relational tuple contains a relation token and multiple argument tokens. Given a binary relation $R$, a relational tuple can be written as $(rel\ arg_1\ arg_2)$ where $arg_1, arg_2$ indicate two arguments of relation $rel$. Let us adopt the two positional roles, $p_i^{rel} = arg_i\text{-}of\text{-}rel$ for $i = 1, 2$. The filler of role $p_i^{rel}$ is $arg_i$. Now let us use role decomposition recursively, noting that the role $p_i^{rel}$ can itself be decomposed into a sub-role $p_i = arg_i\text{-}of\text{-}\_$ which has a sub-filler $rel$. Suppose that $arg_i, rel, p_i$ are embedded as vectors $\boldsymbol{a}_i, \boldsymbol{r}, \boldsymbol{p}_i$. Then the TPR encoding of $p_i^{rel}$ is $\boldsymbol{r} \otimes \boldsymbol{p}_i$, so the TPR encoding of filler $arg_i$ bound to role $p_i^{rel}$ is $\boldsymbol{a}_i \otimes (\boldsymbol{r} \otimes \boldsymbol{p}_i)$. The tensor product is associative, so we can omit parentheses and write the TPR for the formal-language expression, the relational tuple $(rel\ arg_1\ arg_2)$, as:

$$\mathbf{H} = \boldsymbol{a}_1 \otimes \boldsymbol{r} \otimes \boldsymbol{p}_1 + \boldsymbol{a}_2 \otimes \boldsymbol{r} \otimes \boldsymbol{p}_2. \tag{3}$$

Given the unbinding vectors $\boldsymbol{p}_i'$ for positional role vectors $\boldsymbol{p}_i$ and the unbinding vector $\boldsymbol{r}'$ for the vector $\boldsymbol{r}$ that embeds relation $rel$, each argument can be unbound in two steps as shown in Eqs. 4–5.

$$\mathbf{H} \cdot \boldsymbol{p}_i' = [\boldsymbol{a}_1 \otimes \boldsymbol{r} \otimes \boldsymbol{p}_1 + \boldsymbol{a}_2 \otimes \boldsymbol{r} \otimes \boldsymbol{p}_2] \cdot \boldsymbol{p}_i' = \boldsymbol{a}_i \otimes \boldsymbol{r} \tag{4}$$

$$[\boldsymbol{a}_i \otimes \boldsymbol{r}] \cdot \boldsymbol{r}' = \boldsymbol{a}_i \tag{5}$$

Here $\cdot$ denotes the tensor inner product, which for the order-3 $\mathbf{H}$ and order-1 $\boldsymbol{p}_i'$ in Eq. 4 can be defined as $[\mathbf{H} \cdot \boldsymbol{p}_i']_{jk} = \sum_l [\mathbf{H}]_{jkl} [\boldsymbol{p}_i']_l$; in Eq. 5, $\cdot$ is equivalent to the matrix-vector product.

Our proposed scheme can be contrasted with the TPR scheme in which $(rel\ arg_1\ arg_2)$ is embedded as $\boldsymbol{r} \otimes \boldsymbol{a}_1 \otimes \boldsymbol{a}_2$ (e.g., Smolensky et al. (2016); Schlag & Schmidhuber (2018)). In that scheme, an $n$-ary-relation tuple is embedded as an order-$(n+1)$ tensor, and unbinding an argument requires knowing all the other arguments (to use their unbinding vectors). In the scheme proposed here, an $n$-ary-relation tuple is still embedded as an order-3 tensor: there are just $n$ terms in the sum in Eq. 3, using $n$ position vectors $\boldsymbol{p}_1, \ldots, \boldsymbol{p}_n$; unbinding simply requires knowing the unbinding vectors for these fixed position vectors.

In the model, the order-3 tensor $\mathbf{H}$ of Eq. 3 has a different status than the order-2 tensor $\mathbf{T}_S$ of Sec. 3.1.1. $\mathbf{T}_S$ is a TPR by construction, whereas $\mathbf{H}$ is a TPR as a result of successful learning. To generate the output relational tuples, the decoder assumes each tuple has the form of Eq. 3, and performs the unbinding operations which that structure calls for. In Appendix Sec. A.3, it is shown that, if unbinding each of a set of roles from some unknown tensor $\mathbf{T}$ gives a target set of fillers, then $\mathbf{T}$ must equal the TPR generated by those role/filler pairs, plus some tensor that is irrelevant because unbinding from it produces the zero vector. In other words, if the decoder succeeds in producing filler vectors that correspond to output relational tuples that match the target, then, as far as what the decoder can see, the tensor that it operates on is the TPR of Eq. 3.

### 3.1.3 THE TP-N2F SCHEME FOR LEARNING THE INPUT-OUTPUT MAPPING

To generate formal relational tuples from natural-language descriptions, a learning strategy for the mapping between the two structures is particularly important. As shown in (6), we formalize the learning scheme as learning a mapping function $f_{\text{mapping}}(\cdot)$, which, given a structural representation of the natural-language input, $\mathbf{T}_S$, outputs a tensor $\mathbf{T}_F$ from which the structural representation of the output can be generated. At the role level of description, there's nothing more to be said about this mapping; how it is modeled at the neural network level is discussed in Sec. 3.2.1.

$$\mathbf{T}_F = f_{\text{mapping}}(\mathbf{T}_S) \tag{6}$$

## 3.2 THE TP-N2F MODEL FOR NATURAL- TO FORMAL-LANGUAGE GENERATION

As shown in Figure 1, the TP-N2F model is implemented with three steps: encoding, mapping, and decoding. The encoding step is implemented by the TP-N2F natural-language encoder (*TP-N2F Encoder*), which takes the sequence of word tokens as inputs, and encodes them via TPR binding according to the TP-N2F role scheme for natural-language input given in Sec. 3.1.1. The mapping step is implemented by an MLP called the *Reasoning Module*, which takes the encoding produced by the TP-N2F Encoder as input. It learns to map the natural-language-structure encoding of the input to a representation that will be processed under the assumption that it follows the role scheme for output relational-tuples specified in Sec. 3.1.2: the model needs to learn to produce TPRs such that this processing generates correct output programs. The decoding step is implemented by the TP-N2F relational tuples decoder (*TP-N2F Decoder*), which takes the output from the Reasoning Module (Sec. 3.1.3) and decodes the target sequence of relational tuples via TPR unbinding. The TP-N2F Decoder utilizes an attention mechanism over the individual-word TPRs $\mathbf{T}^t$ produced by the TP-N2F Encoder. The detailed implementations are introduced below.

### 3.2.1 THE TP-N2F NATURAL-LANGUAGE ENCODER

The TP-N2F encoder follows the role scheme in Sec. 3.1.1 to encode each word token $w^t$ by soft-selecting one of $n_F$ fillers and one of $n_R$ roles. The fillers and roles are embedded as vectors. These embedding vectors, and the functions for selecting fillers and roles, are learned by two LSTMs, the Filler-LSTM and the Role-LSTM. (See Figure 2.) At each time-step $t$, the Filler-LSTM and the Role-LSTM take a learned word-token embedding $w^t$ as input. The hidden state of the Filler-LSTM, $\boldsymbol{h}_F^t$, is used to compute softmax scores $u_k^F$ over $n_F$ filler slots, and a filler vector $\boldsymbol{f}^t = \boldsymbol{F}\boldsymbol{u}^F$ is computed from the softmax scores (recall from Sec. 2 that $\boldsymbol{F}$ is the learned matrix of filler vectors). Similarly, a role vector is computed from the hidden state of the Role-LSTM, $\boldsymbol{h}_R^t$. $f_F$ and $f_R$ denote the functions that generate $\boldsymbol{f}^t$ and $\boldsymbol{r}^t$ from the hidden states of the two LSTMs. The token $w^t$ is encoded as $\mathbf{T}^t$, the tensor product of $\boldsymbol{f}^t$ and $\boldsymbol{r}^t$. $\mathbf{T}^t$ replaces the hidden vector in each LSTM and is passed to the next time step, together with the LSTM cell-state vector $\boldsymbol{c}^t$: see (7)–(8). After encoding the whole sequence, the TP-N2F encoder outputs the sum of all tensor products $\sum_t \mathbf{T}^t$ to the next module. We use an MLP, called the Reasoning MLP, for TPR mapping; it takes an order-2 TPR from the encoder and maps it to the initial state of the decoder. Detailed equations and implementation are provided in Sec. A.2.1 of the Appendix.

$$\boldsymbol{h}_F^t = f_{\text{Filler}-\text{LSTM}}(\boldsymbol{w}^t, \mathbf{T}^{t-1}, \boldsymbol{c}_F^{t-1}) \qquad \boldsymbol{h}_R^t = f_{\text{Role}-\text{LSTM}}(\boldsymbol{w}^t, \mathbf{T}^{t-1}, \boldsymbol{c}_R^{t-1}) \tag{7}$$

$$\mathbf{T}^t = \boldsymbol{f}^t \otimes \boldsymbol{r}^t = f_F(\boldsymbol{h}_F^t) \otimes f_R(\boldsymbol{h}_R^t) \tag{8}$$

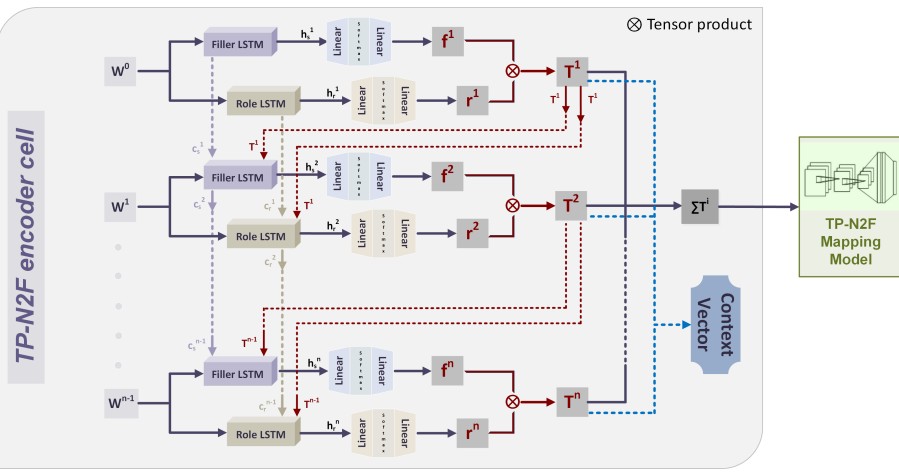

Figure 2: Implementation of the TP-N2F encoder.

### 3.2.2 The TP-N2F Relational-Tuple Decoder

The TP-N2F Decoder is an RNN that takes the output from the reasoning MLP as its initial hidden state for generating a sequence of relational tuples (Figure 3). This decoder contains an attentional LSTM called the Tuple-LSTM which feeds an unbinding module: attention operates on the context vector of the encoder, consisting of all individual encoder outputs $\{\mathbf{T}^t\}$. The hidden-state $\mathbf{H}$ of the Tuple-LSTM is treated as a TPR of a relational tuple and is unbound to a relation and arguments. During training, the Tuple-LSTM needs to learn a way to make $\mathbf{H}$ suitably approximate a TPR. At each time step $t$, the hidden state $\mathbf{H}^t$ of the Tuple-LSTM with attention (The version in Luong et al. (2015)) (9) is fed as input to the unbinding module, which regards $\mathbf{H}^t$ as if it were the TPR of a relational tuple with $m$ arguments possessing the role structure described in Sec. 3.1.2: $\mathbf{H}^t \approx \sum_{i=1}^{m} \boldsymbol{a}_i^t \otimes \boldsymbol{r}^t \otimes \boldsymbol{p}_i$. (In Figure 3, the assumed hypothetical form of $\mathbf{H}^t$, as well as that of $\mathbf{B}_i^t$ below, is shown in a bubble with dashed border.) To decode a binary relational tuple, the unbinding module decodes it from $\mathbf{H}^t$ using the two steps of TPR unbinding given in (4)–(5). The positional unbinding vectors $\boldsymbol{p}_i'$ are learned during training and shared across all time steps. After the first unbinding step (4), i.e., the inner product of $\mathbf{H}^t$ with $\boldsymbol{p}_i'$, we get tensors $\mathbf{B}_i^t$ (10). These are treated as the TPRs of two arguments $\boldsymbol{a}_i^t$ bound to a relation $\boldsymbol{r}^t$. A relational unbinding vector $\boldsymbol{r}'^t$ is computed by a linear function from the sum of the $\mathbf{B}_i^t$ and used to compute the inner product with each $\mathbf{B}_i^t$ to yield $\boldsymbol{a}_i^t$, which are treated as the embedding of argument vectors (11). Based on the TPR theory, $\boldsymbol{r}'^t$ is passed to a linear function to get $\boldsymbol{r}^t$ as the embedding of a relation vector. Finally, the softmax probability distribution over symbolic outputs is computed for relations and arguments separately. In generation, the most probable symbol is selected. (More detailed equations are in Appendix Sec. A.2.3)

$$\mathbf{H}^t = \text{Atten}(f_{\text{Tuple}-\text{LSTM}}(rel^t, arg_1^t, arg_2^t, \mathbf{H}^{t-1}, c^{t-1}), [\mathbf{T}^0, ..., \mathbf{T}^{n-1}]) \tag{9}$$

$$\mathbf{B}_1^t = \mathbf{H}^t \cdot \boldsymbol{p}_1' \qquad \mathbf{B}_2^t = \mathbf{H}^t \cdot \boldsymbol{p}_2' \tag{10}$$

$$\boldsymbol{r}'^t = f_{\text{linear}}(\mathbf{B}_1^t + \mathbf{B}_2^t) \qquad \boldsymbol{a}_1^t = \mathbf{B}_1^t \cdot \boldsymbol{r}'^t \qquad \boldsymbol{a}_2^t = \mathbf{B}_2^t \cdot \boldsymbol{r}'^t \tag{11}$$

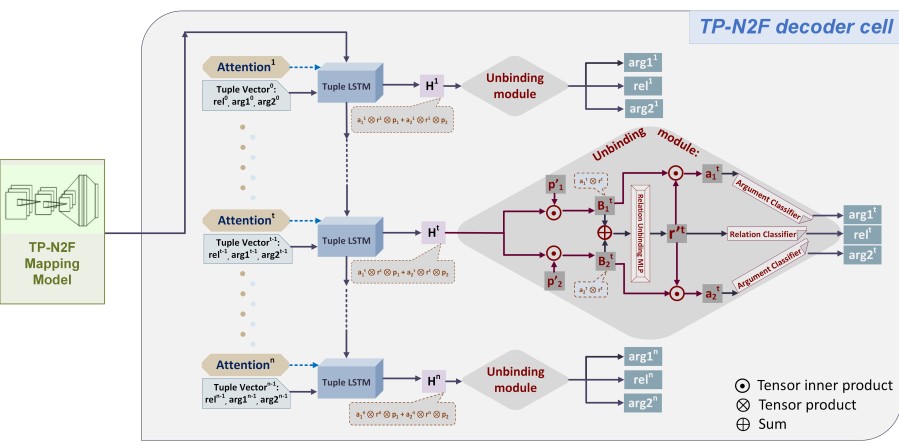

Figure 3: Implementation of the TP-N2F decoder.

### 3.3 Inference and The Learning Strategy of the TP-N2F Model

During inference time, natural language questions are encoded via the encoder and the Reasoning MLP maps the output of the encoder to the input of the decoder. We use greedy decoding (selecting the most likely class) to decode one relation and its arguments. The relation and argument vectors are concatenated to construct a new vector as the input for the Tuple-LSTM in the next step.

TP-N2F is trained using back-propagation (Rumelhart et al., 1986) with the Adam optimizer (Kingma & Ba, 2017) and teacher-forcing. At each time step, the ground-truth relational tuple is provided as the input for the next time step. As the TP-N2F decoder decodes a relational tuple at each time step, the relation token is selected only from the relation vocabulary and the argument tokens from the argument vocabulary. For an input $\mathcal{I}$ that generates $N$ output relational tuples, the loss

is the sum of the cross entropy loss $\mathcal{L}$ between the true labels $L$ and predicted tokens for relations and arguments as shown in (12).

$$\mathcal{L}_{\mathcal{I}} = \sum_{i=0}^{N-1} \mathcal{L}(rel^i, L_{rel^i}) + \sum_{i=0}^{N-1} \sum_{j=1}^{2} \mathcal{L}(arg_j^i, L_{arg_j^i}) \tag{12}$$

## 4 EXPERIMENTS

The proposed TP-N2F model is evaluated on two N2F tasks, generating operation sequences to solve math problems and generating Lisp programs. In both tasks, TP-N2F achieves state-of-the-art performance. We further analyze the behavior of the unbinding relation vectors in the proposed model. Results of each task and the analysis of the unbinding relation vectors are introduced in turn. Details of experiments and datasets are described in Sec. A.1 in the Appendix.

### 4.1 GENERATING OPERATION SEQUENCES TO SOLVE MATH PROBLEMS

Given a natural-language math problem, we need to generate a sequence of operations (operators and corresponding arguments) from a set of operators and arguments to solve the given problem. Each operation is regarded as a relational tuple by viewing the operator as relation, e.g., $(add, n1, n2)$. We test TP-N2F for this task on the MathQA dataset (Amini et al., 2019). The MathQA dataset consists of about 37k math word problems, each with a corresponding list of multi-choice options and the corresponding operation sequence. In this task, TP-N2F is deployed to generate the operation sequence given the question. The generated operations are executed with the execution script from Amini et al. (2019) to select a multi-choice answer. As there are about 30% noisy data (where the execution script returns the wrong answer when given the ground-truth program; see Sec. A.1 of the Appendix), we report both execution accuracy (of the final multi-choice answer after running the execution engine) and operation sequence accuracy (where the generated operation sequence must match the ground truth sequence exactly). TP-N2F is compared to a baseline provided by the seq2prog model in Amini et al. (2019), an LSTM-based seq2seq model with attention. Our model outperforms both the original seq2prog, designated SEQ2PROG-orig, and the best reimplemented seq2prog after an extensive hyperparameter search, designated SEQ2PROG-best. Table 1 presents the results. To verify the importance of the TP-N2F encoder and decoder, we conducted experiments to replace either the encoder with a standard LSTM (denoted LSTM2TP) or the decoder with a standard attentional LSTM (denoted TP2LSTM). We observe that both the TPR components of TP-N2F are important for achieving the observed performance gain relative to the baseline.

Table 1: **Results on MathQA dataset testing set**

| MODEL | Operation Accuracy(%) | Execution Accuracy(%) |
|---|---|---|
| SEQ2PROG-orig | 59.4 | 51.9 |
| SEQ2PROG-best | 66.97 | 54.0 |
| TP2LSTM (ours) | 68.84 | 54.61 |
| LSTM2TP (ours) | 68.21 | 54.61 |
| **TP-N2F** (ours) | **71.89** | **55.95** |

### 4.2 GENERATING PROGRAM TREES FROM NATURAL-LANGUAGE DESCRIPTIONS

Generating Lisp programs requires sensitivity to structural information because Lisp code can be regarded as tree-structured. Given a natural-language query, we need to generate code containing function calls with parameters. Each function call is a relational tuple, which has a function as the relation and parameters as arguments. We evaluate our model on the AlgoLisp dataset for this task and achieve state-of-the-art performance. The AlgoLisp dataset (Polosukhin & Skidanov, 2018) is a program synthesis dataset. Each sample contains a problem description, a corresponding Lisp program tree, and 10 input-output testing pairs. We parse the program tree into a straight-line sequence of tuples (same style as in MathQA). AlgoLisp provides an execution script to run the generated program and has three evaluation metrics: the accuracy of passing all test cases (Acc), the accuracy of passing 50% of test cases (50p-Acc), and the accuracy of generating an exactly matching program (M-Acc). AlgoLisp has about 10% noisy data (details in the Appendix), so we report results both on the full test set and the cleaned test set (in which all noisy testing samples are removed). TP-N2F is

Table 2: **Results of AlgoLisp dataset**

| MODEL (%) | Full Testing Set | | | Cleaned Testing Set | | |
|---|---|---|---|---|---|---|
| | Acc | 50p-Acc | M-Acc | Acc | 50p-Acc | M-Acc |
| Seq2Tree | 61.0 | | | | | |
| LSTM2LSTM+atten | 67.54 | 70.89 | 75.12 | 76.83 | 78.86 | 75.42 |
| TP2LSTM (ours) | 72.28 | 77.62 | 79.92 | 77.67 | 80.51 | 76.75 |
| LSTM2TPR (ours) | 75.31 | 79.26 | 83.05 | 84.44 | 86.13 | 83.43 |
| SAPSpre-VH-Att-256 | 83.80 | 87.45 | | 92.98 | 94.15 | |
| **TP-N2F** (ours) | **84.02** | **88.01** | **93.06** | **93.48** | **94.64** | **92.78** |

compared with an LSTM seq2seq with attention model, the Seq2Tree model in Polosukhin & Skidanov (2018), and a seq2seq model with a pre-trained tree decoder from the Tree2Tree autoencoder (SAPS) reported in Bednarek et al. (2019). As shown in Table 2, TP-N2F outperforms all existing models on both the full test set and the cleaned test set. Ablation experiments with TP2LSTM and LSTM2TP show that, for this task, the TP-N2F Decoder is more helpful than TP-N2F Encoder. This may be because lisp codes rely more heavily on structure representations.

### 4.3 INTERPRETATION OF LEARNED STRUCTURE

To interpret the structure learned by the model, we extract the trained unbinding relation vectors from the TP-N2F Decoder and reduce the dimension of vectors via Principal Component Analysis. K-means clustering results on the average vectors are presented in Figure 4 and Figure 5 (in Appendix A.6). Results show that unbinding vectors for operators or functions with similar semantics tend to be close to each other. For example, with 5 clusters in the MathQA dataset, arithmetic operators such as *add, subtract, multiply, divide* are clustered together, and operators related to *square* or *volume* of geometry are clustered together. With 4 clusters in the AlgoLisp dataset, partial/lambda functions and sort functions are in one cluster, and string processing functions are clustered together. Note that there is no direct supervision to inform the model about the nature of the operations, and the TP-N2F decoder has induced this role structure using weak supervision signals from question/operation-sequence-answer pairs. More clustering results are presented in the Appendix A.6.

## 5 RELATED WORK

N2F tasks include many different subtasks such as symbolic reasoning or semantic parsing (Kamath & Das, 2019; Cai & Lam, 2019; Liao et al., 2018; Amini et al., 2019; Polosukhin & Skidanov, 2018; Bednarek et al., 2019). These tasks require models with strong structure-learning ability. TPR is a promising technique for encoding symbolic structural information and modeling symbolic reasoning in vector space. TPR binding has been used for encoding and exploring grammatical structural information of natural language (Palangi et al., 2018; Huang et al., 2019). TPR unbinding has also been used to generate natural language captions from images (Huang et al., 2018). Some researchers use TPRs for modeling deductive reasoning processes both on a rule-based model and deep learning models in vector space (Lee et al., 2016; Smolensky et al., 2016; Schlag & Schmidhuber, 2018). However, none of these previous models takes advantage of combining TPR binding and TPR unbinding to learn structure representation mappings explicitly, as done in our model. Although researchers are paying increasing attention to N2F tasks, most of the proposed models either do not encode structural information explicitly or are specialized to particular tasks. Our proposed TP-N2F neural model can be applied to many tasks.

## 6 CONCLUSION AND FUTURE WORK

In this paper we propose a new scheme for neural-symbolic relational representations and a new architecture, TP-N2F, for formal-language generation from natural-language descriptions. To our knowledge, TP-N2F is the first model that combines TPR binding and TPR unbinding in the encoder-decoder fashion. TP-N2F achieves the state-of-the-art on two instances of N2F tasks, showing significant structure learning ability. The results show that both the TP-N2F encoder and the TP-N2F decoder are important for improving natural- to formal-language generation. We believe that the interpretation and symbolic structure encoding of TPRs are a promising direction for future work. We also plan to combine large-scale deep learning models such as BERT with TP-N2F to take advantage of structure learning for other generation tasks.

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

## A  APPENDIX

### A.1  IMPLEMENTATIONS OF TP-N2F FOR EXPERIMENTS

In this section, we present details of the experiments of TP-N2F on the two datasets. We present the implementation of TP-N2F on each dataset.

The MathQA dataset consists of about 37k math word problems ((80/12/8)% training/dev/testing problems), each with a corresponding list of multi-choice options and an straight-line operation sequence program to solve the problem. An example from the dataset is presented in the Appendix A.4. In this task, TP-N2F is deployed to generate the operation sequence given the question. The generated operations are executed to generate the solution for the given math problem. We use the execution script from Amini et al. (2019) to execute the generated operation sequence and compute the multi-choice accuracy for each problem. During our experiments we observed that there are about 30% noisy examples (on which the execution script fails to get the correct answer on the ground truth program). Therefore, we report both execution accuracy (the final multi-choice answer after running the execution engine) and operation sequence accuracy (where the generated operation sequence must match the ground truth sequence exactly).

The AlgoLisp dataset (Polosukhin & Skidanov, 2018) is a program synthesis dataset, which has 79k/9k/10k training/dev/testing samples. Each sample contains a problem description, a corresponding Lisp program tree, and 10 input-output testing pairs. We parse the program tree into a straight-line sequence of commands from leaves to root and (as in MathQA) use the symbol $\#_i$ to indicate the result of the $i^{\text{th}}$ command (generated previously by the model). A dataset sample with our parsed command sequence is presented in the Appendix A.4. AlgoLisp provides an execution script to run the generated program and has three evaluation metrics: accuracy of passing all test cases (Acc), accuracy of passing 50% of test cases (50p-Acc), and accuracy of generating an exactly matched program (M-Acc). AlgoLisp has about 10% noise data (where the execution script fails to pass all test cases on the ground truth program), so we report results both on the full test set and the cleaned test set (in which all noisy testing samples are removed).

We use $d_{\text{R}}, n_{\text{R}}, d_{\text{F}}, n_{\text{F}}$ to indicate the TP-N2F encoder hyperparameters, the dimension of role vectors, the number of roles, the dimension of filler vectors and the number of fillers. $d_{Rel}, d_{Arg}, d_{Pos}$ indicate the TP-N2F decoder hyper-parameters, the dimension of relation vectors, the dimension of argument vectors, and the dimension of position vectors.

In the experiment on the MathQA dataset, we use $n_{\text{F}} = 150$, $n_{\text{R}} = 50$, $d_{\text{F}} = 30$, $d_{\text{R}} = 20$, $d_{Rel} = 20$, $d_{Arg} = 10$, $d_{Pos} = 5$ and we train the model for 60 epochs with learning rate 0.00115. The reasoning module only contains one layer. As most of the math operators in this dataset are binary, we replace all operators taking three arguments with a set of binary operators based on hand-encoded rules, and for all operators taking one argument, a padding symbol is appended. For the baseline SEQ2PROG-orig, TP2LSTM and LSTM2TP, we use hidden size 100, single-direction, one-layer LSTM. For the SEQ2PROG-best, we performed a hyperparameter search on the hidden size for both encoder and decoder; the best score is reported.

In the experiment on the AlgoLisp dataset, we use $n_{\text{F}} = 150$, $n_{\text{R}} = 50$, $d_{\text{F}} = 30$, $d_{\text{R}} = 30$, $d_{Rel} = 30$, $d_{Arg} = 20$, $d_{Pos} = 5$ and we train the model for 50 epochs with learning rate 0.00115. We also use one-layer in the reasoning module like in MathQA. For this dataset, most function calls take three arguments so we simply add padding symbols for those functions with fewer than three arguments.

## A.2 Detailed equations of TP-N2F

### A.2.1 TP-N2F encoder

**Filler-LSTM in TP-N2F encoder**

This is a standard LSTM, governed by the equations:

$$\boldsymbol{f}_{\mathrm{f}}^t = \varphi(\boldsymbol{U}_{\mathrm{ff}}\,\boldsymbol{w}^t + \boldsymbol{V}_{\mathrm{ff}}\,\flat(\mathbf{T}^{t-1}) + \boldsymbol{b}_{\mathrm{ff}}) \tag{13}$$

$$\boldsymbol{g}_{\mathrm{f}}^t = \tanh(\boldsymbol{U}_{\mathrm{fg}}\,\boldsymbol{w}^t + \boldsymbol{V}_{\mathrm{fg}}\,\flat(\mathbf{T}^{t-1}) + \boldsymbol{b}_{\mathrm{fg}}) \tag{14}$$

$$\boldsymbol{i}_{\mathrm{f}}^t = \varphi(\boldsymbol{U}_{\mathrm{fi}}\,\boldsymbol{w}^t + \boldsymbol{V}_{\mathrm{fi}}\,\flat(\mathbf{T}^{t-1}) + \boldsymbol{b}_{\mathrm{fi}}) \tag{15}$$

$$\boldsymbol{o}_{\mathrm{f}}^t = \varphi(\boldsymbol{U}_{\mathrm{fo}}\,\boldsymbol{w}^t + \boldsymbol{V}_{\mathrm{fo}}\,\flat(\mathbf{T}^{t-1}) + \boldsymbol{b}_{\mathrm{fo}}) \tag{16}$$

$$\boldsymbol{c}_{\mathrm{f}}^t = \boldsymbol{f}_{\mathrm{f}}^t \odot \boldsymbol{c}_{\mathrm{f}}^{t-1} + \boldsymbol{i}_{\mathrm{f}}^t \odot \boldsymbol{g}_{\mathrm{f}}^t \tag{17}$$

$$\boldsymbol{h}_{\mathrm{f}}^t = \boldsymbol{o}_{\mathrm{f}}^t \odot \tanh(\boldsymbol{c}_f^t) \tag{18}$$

$\varphi, \tanh$ are the logistic sigmoid and tanh functions applied elementwise. $\flat$ flattens (reshapes) a matrix in $\mathbb{R}^{d_{\mathrm{F}} \times d_{\mathrm{R}}}$ into a vector in $\mathbb{R}^{d_{\mathrm{T}}}$, where $d_{\mathrm{T}} = d_{\mathrm{F}} d_{\mathrm{R}}$. $\odot$ is elementwise multiplication. The variables have the following dimensions:

$$\boldsymbol{f}_{\mathrm{f}}^t, \boldsymbol{g}_{\mathrm{f}}^t, \boldsymbol{i}_{\mathrm{f}}^t, \boldsymbol{o}_{\mathrm{f}}^t, \boldsymbol{c}_{\mathrm{f}}^t, \boldsymbol{h}_{\mathrm{f}}^t, \boldsymbol{b}_{\mathrm{ff}}, \boldsymbol{b}_{\mathrm{fg}}, \boldsymbol{b}_{\mathrm{fi}}, \boldsymbol{b}_{\mathrm{fo}}, \flat(\mathbf{T}^{t-1}) \in \mathbb{R}^{d_{\mathrm{T}}}$$

$$w^t \in \mathbb{R}^d$$

$$\boldsymbol{U}_{\mathrm{ff}}, \boldsymbol{U}_{\mathrm{fg}}, \boldsymbol{U}_{\mathrm{fi}}, \boldsymbol{U}_{\mathrm{fo}} \in \mathbb{R}^{d_{\mathrm{T}} \times d}$$

$$\boldsymbol{V}_{\mathrm{ff}}, \boldsymbol{V}_{\mathrm{fg}}, \boldsymbol{V}_{\mathrm{fi}}, \boldsymbol{V}_{\mathrm{fo}} \in \mathbb{R}^{d_{\mathrm{T}} \times d_{\mathrm{T}}}$$

**Filler vector**

The filler vector for input token $w^t$ is $\boldsymbol{f}^t$, defined through an attention vector over possible fillers, $\boldsymbol{a}_{\mathrm{f}}^t$:

$$\boldsymbol{a}_{\mathrm{f}}^t = \mathrm{softmax}((\boldsymbol{W}_{\mathrm{fa}}\,\boldsymbol{h}_{\mathrm{f}}^t)/T) \tag{19}$$

$$\boldsymbol{f}^t = \boldsymbol{W}_{\mathrm{f}}\,\boldsymbol{a}_{\mathrm{f}}^t \tag{20}$$

($W_{\mathrm{f}}$ is the same as $\boldsymbol{F}$ of Sec. 2.) The variables' dimensions are:

$$\boldsymbol{W}_{\mathrm{fa}} \in \mathbb{R}^{n_{\mathrm{F}} \times d_{\mathrm{T}}}$$

$$\boldsymbol{a}_{\mathrm{f}}^t \in \mathbb{R}^{n_{\mathrm{F}}}$$

$$\boldsymbol{W}_{\mathrm{f}} \in \mathbb{R}^{d_{\mathrm{F}} \times n_{\mathrm{F}}}$$

$$\boldsymbol{f}^t \in \mathbb{R}^{d_{\mathrm{F}}}$$

$T$ is the temperature factor, which is fixed at 0.1.

**Role-LSTM in TP-N2F encoder**

Similar to the Filler-LSTM, the Role-LSTM is also a standard LSTM, governed by the equations:

$$\boldsymbol{f}_{\mathrm{r}}^t = \varphi(\boldsymbol{U}_{\mathrm{rf}}\,\boldsymbol{w}^t + \boldsymbol{V}_{\mathrm{rf}}\,\flat(\mathbf{T}^{t-1}) + \boldsymbol{b}_{\mathrm{rf}}) \tag{21}$$

$$\boldsymbol{g}_{\mathrm{r}}^t = \tanh(\boldsymbol{U}_{\mathrm{rg}}\,\boldsymbol{w}^t + \boldsymbol{V}_{\mathrm{rg}}\,\flat(\mathbf{T}^{t-1}) + \boldsymbol{b}_{\mathrm{rg}}) \tag{22}$$

$$\boldsymbol{i}_{\mathrm{r}}^t = \varphi(\boldsymbol{U}_{\mathrm{ri}}\,\boldsymbol{w}^t + \boldsymbol{V}_{\mathrm{ri}}\,\flat(\mathbf{T}^{t-1}) + \boldsymbol{b}_{\mathrm{ri}}) \tag{23}$$

$$\boldsymbol{o}_{\mathrm{r}}^t = \varphi(\boldsymbol{U}_{\mathrm{ro}}\,\boldsymbol{w}^t + \boldsymbol{V}_{\mathrm{ro}}\,\flat(\mathbf{T}^{t-1}) + \boldsymbol{b}_{\mathrm{ro}}) \tag{24}$$

$$\boldsymbol{c}_{\mathrm{r}}^t = \boldsymbol{f}_{\mathrm{r}}^t \odot \boldsymbol{c}_{\mathrm{r}}^{t-1} + \boldsymbol{i}_{\mathrm{r}}^t \odot \boldsymbol{g}_{\mathrm{r}}^t \tag{25}$$

$$\boldsymbol{h}_{\mathrm{r}}^t = \boldsymbol{o}_{\mathrm{r}}^t \odot \tanh(\boldsymbol{c}_{\mathrm{r}}^t) \tag{26}$$

The variable dimensions are:

$$\boldsymbol{f}_{\mathrm{r}}^t, \boldsymbol{g}_{\mathrm{r}}^t, \boldsymbol{i}_{\mathrm{r}}^t, \boldsymbol{o}_{\mathrm{r}}^t, \boldsymbol{c}_{\mathrm{r}}^t, \boldsymbol{h}_{\mathrm{r}}^t, \boldsymbol{b}_{\mathrm{rf}}, \boldsymbol{b}_{\mathrm{rg}}, \boldsymbol{b}_{\mathrm{ri}}, \boldsymbol{b}_{\mathrm{ro}}, \flat(\mathbf{T}^{t-1}) \in \mathbb{R}^{d_{\mathrm{T}}}$$

$$w^t \in \mathbb{R}^d$$

$$\boldsymbol{U}_{\mathrm{rf}}, \boldsymbol{U}_{\mathrm{rg}}, \boldsymbol{U}_{\mathrm{ri}}, \boldsymbol{U}_{\mathrm{ro}} \in \mathbb{R}^{d_{\mathrm{T}} \times d}$$

$$\boldsymbol{V}_{\mathrm{rf}}, \boldsymbol{V}_{\mathrm{rg}}, \boldsymbol{V}_{\mathrm{ri}}, \boldsymbol{V}_{\mathrm{ro}} \in \mathbb{R}^{d_{\mathrm{T}} \times d_{\mathrm{T}}}$$

**Role vector**

The role vector for input token $w^t$ is determined analogously to its filler vector:

$$\boldsymbol{a}_{\text{r}}^t = \text{softmax}((\boldsymbol{W}_{\text{ra}}\,\boldsymbol{h}_{\text{r}}^t)/T) \tag{27}$$

$$\boldsymbol{r}^t = \boldsymbol{W}_{\text{r}}\,\boldsymbol{a}_{\text{r}}^t \tag{28}$$

The dimensions are:

$$\boldsymbol{W}_{\text{ra}} \in \mathbb{R}^{n_{\text{R}} \times d_{\text{T}}}$$
$$\boldsymbol{a}_{\text{r}}^t \in \mathbb{R}^{n_{\text{R}}}$$
$$\boldsymbol{W}_{\text{r}} \in \mathbb{R}^{d_{\text{R}} \times n_{\text{R}}}$$
$$\boldsymbol{r}^t \in \mathbb{R}^{d_{\text{R}}}$$

**Binding**

The TPR for the filler/role binding for token $w^t$ is then:

$$\boldsymbol{T}_t = \boldsymbol{r}^t \otimes \boldsymbol{f}^t \tag{29}$$

where

$$\boldsymbol{T}^t \in \mathbb{R}^{d_{\text{R}} \times d_{\text{F}}}$$

### A.2.2 STRUCTURE MAPPING

$$\mathsf{H}^0 = f_{\text{mapping}}(\boldsymbol{T}_t) \tag{30}$$

$\mathsf{H}^0 \in \mathbb{R}^{d_{\text{H}}}$, where $d_{\text{H}} = d_{\text{A}}, d_{\text{O}}, d_{\text{P}}$ are dimension of argument vector, operator vector and position vector. $f_{\text{mapping}}$ is implemented with a MLP (linear layer followed by a tanh) for mapping the $\boldsymbol{T}_t \in \mathbb{R}^{d_{\text{T}}}$ to the initial state of decoder $\mathsf{H}^0$.

### A.2.3 TP-N2F DECODER

**Tuple-LSTM**

The output tuples are also generated via a standard LSTM:

$$\boldsymbol{w}_d^t = \gamma(\boldsymbol{w}_{Rel}^{t-1}, \boldsymbol{w}_{Arg1}^{t-1}, \boldsymbol{w}_{Arg2}^{t-1}) \tag{31}$$

$$\boldsymbol{f}^t = \varphi(\boldsymbol{U}_{\text{f}}\,\boldsymbol{w}_d^t + \boldsymbol{V}_{\text{f}}\,\flat(\mathsf{H}^{t-1}) + \boldsymbol{b}_{\text{f}}) \tag{32}$$

$$\boldsymbol{g}^t = \tanh(\boldsymbol{U}_{\text{g}}\,\boldsymbol{w}_d^t + \boldsymbol{V}_{\text{g}}\,\flat(\mathsf{H}^{t-1}) + \boldsymbol{b}_{\text{g}}) \tag{33}$$

$$\boldsymbol{i}^t = \varphi(\boldsymbol{U}_{\text{i}}\,\boldsymbol{w}_d^t + \boldsymbol{V}_{\text{i}}\,\flat(\mathsf{H}^{t-1}) + \boldsymbol{b}_{\text{i}}) \tag{34}$$

$$\boldsymbol{o}^t = \varphi(\boldsymbol{U}_{\text{o}}\,\boldsymbol{w}_d^t + \boldsymbol{V}_{\text{o}}\,\flat(\mathsf{H}^{t-1}) + \boldsymbol{b}_{\text{o}}) \tag{35}$$

$$\boldsymbol{c}^t = \boldsymbol{f}^t \odot \boldsymbol{c}^{t-1} + \boldsymbol{i}^t \odot \boldsymbol{g}^t \tag{36}$$

$$\boldsymbol{h}_{\text{input}}^t = \boldsymbol{o}^t \odot \tanh(\boldsymbol{c}^t) \tag{37}$$

$$\mathsf{H}^t = \text{Atten}(\boldsymbol{h}_{\text{input}}^t, [\boldsymbol{T}_0, ..., \boldsymbol{T}_{n-1}]) \tag{38}$$

Here, $\gamma$ is the concatenation function. $\boldsymbol{w}_{Rel}^{t-1}$ is the trained embedding vector for the Relation of the input binary tuple, $\boldsymbol{w}_{Arg1}^{t-1}$ is the embedding vector for the first argument and $\boldsymbol{w}_{Arg2}^{t-1}$ is the embedding vector for the second argument. Then the input for the Tuple LSTM is the concatenation of the embedding vectors of relation and arguments, with dimension $d_{\text{dec}}$.

$$\boldsymbol{f}^t, \boldsymbol{g}^t, \boldsymbol{i}^t, \boldsymbol{o}^t, \boldsymbol{c}^t, \boldsymbol{h}_{\text{input}}^t, \boldsymbol{b}_{\text{f}}, \boldsymbol{b}_{\text{g}}, \boldsymbol{b}_{\text{i}}, \boldsymbol{b}_{\text{o}}, \flat(\mathsf{H}^{t-1}) \in \mathbb{R}^{d_{\text{H}}}$$
$$\boldsymbol{w}_d^t \in \mathbb{R}^{d_{\text{dec}}}$$
$$\boldsymbol{U}_{\text{f}}, \boldsymbol{U}_{\text{g}}, \boldsymbol{U}_{\text{i}}, \boldsymbol{U}_{\text{o}} \in \mathbb{R}^{d_{\text{H}} \times d_{\text{dec}}}$$
$$\boldsymbol{V}_{\text{f}}, \boldsymbol{V}_{\text{g}}, \boldsymbol{V}_{\text{i}}, \boldsymbol{V}_{\text{o}} \in \mathbb{R}^{d_{\text{H}} \times d_{\text{H}}}$$
$$\mathsf{H}^t \in \mathbb{R}^{d_{\text{H}}}$$

Atten is the attention mechanism used in Luong et al. (2015), which computes the dot product between $h_{\text{input}}^t$ and each $T_{t'}$. Then a linear function is used on the concatenation of $h_{\text{input}}^t$ and the softmax scores on all dot products to generate $\mathbf{H}^t$. The following equations show the attention mechanism:

$$d^t = \text{score}(h_{\text{input}}^t, \mathbf{C}_T) \tag{39}$$

$$s^t = \mathbf{C}_T \, \text{softmax}(d^t) \tag{40}$$

$$\mathbf{H}^t = K\gamma(h_{\text{input}}^t, s^t) \tag{41}$$

score is the score function of the attention. In this paper, the score function is dot product.

$$\mathbf{C}_T \in \mathbb{R}^{d_{\text{H}} \times n}$$

$$d_t \in \mathbb{R}^n$$

$$s_t \in \mathbb{R}^{d_{\text{H}}}$$

$$K \in \mathbb{R}^{d_{\text{H}} \times (d_{\text{T}}+n)}$$

**Unbinding**

At each timestep $t$, the 2-step unbinding process described in Sec. 3.1.2 operates first on an encoding of the triple as a whole, $\mathbf{H}$, using two unbinding vectors $p_i'$ that are learned but fixed for all tuples. This first unbinding gives an encoding of the two operator-argument bindings, $\mathbf{B}_i$. The second unbinding operates on the $\mathbf{B}_i$, using a generated unbinding vector for the operator, $r'$, giving encodings of the arguments, $a_i$. The generated unbinding vector for the operator, $r'$, and the generated encodings of the arguments, $a_i$, each produce a probability distribution over symbolic operator outputs $Rel$ and symbolic argument outputs $Arg_i$; these probabilities are used in the cross-entropy loss function. For generating a single symbolic output, the most-probable symbols are selected.

$$\mathbf{B}_1^t = \mathbf{H}^t \, p_1' \tag{42}$$

$$\mathbf{B}_2^t = \mathbf{H}^t \, p_2' \tag{43}$$

$$r'^t = \mathbf{W}_{\text{dual}} \left( B_1^t + B_2^t \right) \tag{44}$$

$$a_1^t = \mathbf{B}_1^t \, r'^t \tag{45}$$

$$a_2^t = \mathbf{B}_2^t \, r'^t \tag{46}$$

$$l_r^t = \mathbf{L}_r^t \, r'^t \tag{47}$$

$$l_{a_1}^t = \mathbf{L}_a^t \, a_1^t \tag{48}$$

$$l_{a_2}^t = \mathbf{L}_a^t \, a_2^t \tag{49}$$

$$Rel^t = \text{argmax}(\text{softmax}(l_r^t)) \tag{50}$$

$$Arg1^t = \text{argmax}(\text{softmax}(l_{a_1}^t)) \tag{51}$$

$$Arg2^t = \text{argmax}(\text{softmax}(l_{a_2}^t)) \tag{52}$$

The dimensions are:

$$r'^t \in \mathbb{R}^{d_{\text{O}}}$$

$$a_1^t, a_2^t \in \mathbb{R}^{d_{\text{A}}}$$

$$p_1', p_2' \in \mathbb{R}^{d_{\text{P}}}$$

$$\mathbf{B}_1^t, \mathbf{B}_2^t \in \mathbb{R}^{d_{\text{A}} \times d_{\text{O}}}$$

$$\mathbf{W}_{\text{dual}} \in \mathbb{R}^{d_{\text{H}}}$$

$$\mathbf{L}_r^t \in \mathbb{R}^{n_{\text{O}} \times d_{\text{O}}}$$

$$\mathbf{L}_a^t \in \mathbb{R}^{n_{\text{A}} \times d_{\text{A}}}$$

$$l_r^t \in \mathbb{R}^{n_{\text{R}}}$$

$$l_{a_1}^t, l_{a_2}^t \in \mathbb{R}^{n_{\text{A}}}$$

A.3   THE TENSOR THAT IS INPUT TO THE DECODER'S UNBINDING MODULE IS A TPR

Here we show that, if learning is successful, the order-3 tensor $\mathsf{H}$ that each iteration of the decoder's Tuple LSTM feeds to the decoder's Unbinding Module (Figure 3) will be a TPR of the form assumed in Eq. 3, repeated here:

$$\mathsf{H} = \sum_j \boldsymbol{a}_j \otimes \boldsymbol{r} \otimes \boldsymbol{p}_j. \tag{53}$$

The operations performed by the decoder are given in Eqs. 4–5, and Eqs. 10–11, rewritten here:

$$\mathsf{H} \cdot \boldsymbol{p}_i' = \boldsymbol{q}_i \tag{54}$$

$$\boldsymbol{q}_i \cdot \boldsymbol{r}' = \boldsymbol{a}_i \tag{55}$$

This is the standard TPR unbinding operation, used recursively: first with the unbinding vectors for positions, $\boldsymbol{p}_i'$, then with the unbinding vector for the operator, $\boldsymbol{r}'$. It therefore suffices to analyze a single unbinding; the result can then be used recursively. This in effect reduces the problem to the order-2 case. What we will show is: given a set of unbinding vectors $\{\boldsymbol{r}_i'\}$ which are dual to a set of role vectors $\{\boldsymbol{r}_i\}$, with $i$ ranging over some index set $I$, if $\mathsf{H}$ is an order-2 tensor such that

$$\mathsf{H} \cdot \boldsymbol{r}_i' = \boldsymbol{f}_i, \forall i \in I \tag{56}$$

then

$$\mathsf{H} = \sum_{i \in I} \boldsymbol{f}_i \boldsymbol{r}_i^\top + \mathsf{Z} \equiv \mathsf{H}_{\mathrm{TPR}} + \mathsf{Z} \tag{57}$$

for some tensor $\mathsf{Z}$ that annihilates all the unbinding vectors:

$$\mathsf{Z} \cdot \boldsymbol{r}_i' = \boldsymbol{0}, \forall i \in I. \tag{58}$$

If learning is successful, the processing in the decoder will generate the target relational tuple $(R, A_1, A_2)$ by obeying Eq. 54 in the first unbinding, where we have $\boldsymbol{r}_i' = \boldsymbol{p}_i', \boldsymbol{f}_i = \boldsymbol{q}_i, I = \{1, 2\}$, and obeying Eq. 55 in the second unbinding, where we have $\boldsymbol{r}_i' = \boldsymbol{r}', \boldsymbol{f}_i' = \boldsymbol{a}_i$, with $I =$ the set containing only the null index.

Treat rank-2 tensors as matrices; then unbinding is simply matrix-vector multiplication. Assume the set of unbinding vectors is linearly independent (otherwise there would in general be no way to satisfy Eq. 56 exactly, contrary to assumption). Then expand the set of unbinding vectors, if necessary, into a basis $\{\boldsymbol{r}_k'\}_{k \in K \supseteq I}$. Find the dual basis, with $\boldsymbol{r}_k$ dual to $\boldsymbol{r}_k'$ (so that $\boldsymbol{r}_l^\top \boldsymbol{r}_j' = \delta_{lj}$). Because $\{\boldsymbol{r}_k'\}_{k \in K}$ is a basis, so is $\{\boldsymbol{r}_k\}_{k \in K}$, so any matrix $\mathsf{H}$ can be expanded as $\mathsf{H} = \sum_{k \in K} \boldsymbol{v}_k \boldsymbol{r}_k^\top$. Since $\mathsf{H} \boldsymbol{r}_i' = \boldsymbol{f}_i, \forall i \in I$ are the unbinding conditions (Eq. 56), we must have $\boldsymbol{v}_i = \boldsymbol{f}_i, i \in I$. Let $\mathsf{H}_{\mathrm{TPR}} \equiv \sum_{i \in I} \boldsymbol{f}_i \boldsymbol{r}_i^\top$. This is the desired TPR, with fillers $\boldsymbol{f}_i$ bound to the role vectors $\boldsymbol{r}_i$ which are the duals of the unbinding vectors $\boldsymbol{r}_i'$ ($i \in I$). Then we have $\mathsf{H} = \mathsf{H}_{\mathrm{TPR}} + \mathsf{Z}$ (Eq. 57) where $\mathsf{Z} \equiv \sum_{j \in K, j \notin I} \boldsymbol{v}_j \boldsymbol{r}_j^\top$; so $\mathsf{Z} \boldsymbol{r}_i' = \boldsymbol{0}, i \in I$ (Eq. 58). Thus, if training is successful, the model must have learned how to feed the decoder with order-3 TPRs with the structure posited in Eq. 53.

The argument so far addresses the case where the unbinding vectors are linearly independent, making it possible to satisfy Eq. 56 exactly. In relatively high-dimensional vector spaces, it will often happen that even when the number of unbinding vectors exceeds the dimension of their space by a factor of 2 or 3 (which applies to the TP-N2F models presented here), there is a set of role vectors $\{\boldsymbol{r}_k\}_{k \in K}$ approximately dual to $\{\boldsymbol{r}_k'\}_{k \in K}$, such that $\boldsymbol{r}_l^\top \boldsymbol{r}_j' = \delta_{lj}\ \forall l, j \in K$ holds to a good approximation. (If the distribution of normalized unbinding vectors is approximately uniform on the unit sphere, then choosing the approximate dual vectors to equal the unbinding vectors themselves will do, since they will be nearly orthonormal (Anonymous, in prep.). If the $\{\boldsymbol{r}_k'\}_{k \in K}$ are not normalized, we just rescale the role vectors, choosing $\boldsymbol{r}_k = \boldsymbol{r}_k' / \|\boldsymbol{r}_k'\|^2$.) When the number of such role vectors exceeds the dimension of the embedding space, they will be overcomplete, so while it is still true that any matrix $\mathsf{H}$ can be expanded as above ($\mathsf{H} = \sum_{k \in K} \boldsymbol{v}_k \boldsymbol{r}_k^\top$), this expansion will no longer be unique. So while it remains true that $\mathsf{H}$ a TPR, it is no longer uniquely decomposable into filler/role pairs. The claim above does not claim uniqueness in this sense, and remains true.)

### A.4.1   DATA SAMPLE FROM MATHQA DATASET

**Problem**: The present polulation of a town is 3888. Population increase rate is 20%. Find the population of town after 1 year?
**Options**: a) 2500, b) 2100, c) 3500, d) 3600, e) 2700
**Operations**: multiply(n0,n1), divide(#0,const-100), add(n0,#1)

### A.4.2   DATA SAMPLE FROM ALGOLISP DATASET

**Problem**: Consider an array of numbers and a number, decrements each element in the given array by the given number, what is the given array?
**Program Nested List**: (map a (partial1 b –))
**Command-Sequence**: (partial1 b –), (map a #0)

## A.5   GENERATED PROGRAMS COMPARISON

In this section, we display some generated samples from the two datasets, where the TP-N2F model generates correct programs but LSTM-Seq2Seq does not.

**Question**: A train running at the speed of 50 km per hour crosses a post in 4 seconds. What is the length of the train?
**TP-N2F(correct)**:
(multiply,n0,const1000) (divide,#0,const3600) (multiply,n1,#1)
**LSTM(wrong)**:
(multiply,n0,const0.2778) (multiply,n1,#0)

**Question**: 20 is subtracted from 60 percent of a number, the result is 88. Find the number?
**TP-N2F(correct)**:
(add,n0,n2) (divide,n1,const100) (divide,#0,#1)
**LSTM(wrong)**:
(add,n0,n2) (divide,n1,const100) (divide,#0,#1) (multiply,#2,n3) (subtract,#3,n0)

**Question**: The population of a village is 14300. It increases annually at the rate of 15 percent. What will be its population after 2 years?
**TP-N2F(correct)**:
(divide,n1,const100) (add,#0,const1) (power,#1,n2) (multiply,n0,#2)
**LSTM(wrong)**:
(multiply,const4,const100) (sqrt,#0)

**Question**: There are two groups of students in the sixth grade. There are 45 students in group a, and 55 students in group b. If, on a particular day, 20 percent of the students in group a forget their homework, and 40 percent of the students in group b forget their homework, then what percentage of the sixth graders forgot their homework?
**TP-N2F(correct)**:
(add,n0,n1) (multiply,n0,n2) (multiply,n1,n3) (divide,#1,const100) (divide,#2,const100) (add,#3,#4) (divide,#5,#0) (multiply,#6,const100)
**LSTM(wrong)**:
(multiply,n0,n1) (subtract,n0,n1) (divide,#0,#1)

**Question**: 1 divided by 0.05 is equal to
**TP-N2F(correct)**:
(divide,n0,n1)
**LSTM(wrong)**:

(divide,n0,n1) (multiply,n2,#0)

**Question**: Consider a number a, compute factorial of a
**TP-N2F(correct)**:
( ¡=,arg1,1 ) ( -,arg1,1 ) ( self,#1 ) ( *,#2,arg1 ) ( if,#0,1,#3 ) ( lambda1,#4 ) ( invoke1,#5,a )
**LSTM(wrong)**:
( ¡=,arg1,1 ) ( -,arg1,1 ) ( self,#1 ) ( *,#2,arg1 ) ( if,#0,1,#3 ) ( lambda1,#4 ) ( len,a ) ( invoke1,#5,#6
)

**Question**: Given an array of numbers and numbers b and c, add c to elements of the product of
elements of the given array and b, what is the product of elements of the given array and b?
**TP-N2F(correct)**:
( partial, b,* ) ( partial1,c,+ ) ( map,a,#0 ) ( map,#2,#1 )
**LSTM(wrong)**:
( partial1,b,+ ) ( partial1,c,+ ) ( map,a,#0 ) ( map,#2,#1 )

**Question**: You are given an array of numbers a and numbers b , c and d , let how many times you
can replace the median in a with sum of its digits before it becomes a single digit number and b be
the coordinates of one end and c and d be the coordinates of another end of segment e , your task is
to find the length of segment e rounded down
**TP-N2F(correct)**:
( digits arg1 ) ( len #0 ) ( == #1 1 ) ( digits arg1 ) ( reduce #3 0 + ) ( self #4 ) ( + 1 #5 ) ( if #2 0 #6
) ( lambda1 #7 ) ( sort a ) ( len a ) ( / #10 2 ) ( deref #9 #11 ) ( invoke1 #8 #12 ) ( - #13 c ) ( digits
arg1 ) ( len #15 ) ( == #16 1 ) ( digits arg1 ) ( reduce #18 0 + ) ( self #19 ) ( + 1 #20 ) ( if #17 0 #21
) ( lambda1 #22 ) ( sort a ) ( len a ) ( / #25 2 ) ( deref #24 #26 ) ( invoke1 #23 #27 ) ( - #28 c ) ( *
#14 #29 ) ( - b d ) ( - b d ) ( * #31 #32 ) ( + #30 #33 ) ( sqrt #34 ) ( floor #35 )
**LSTM(wrong)**: ( digits arg1 ) ( len #0 ) ( == #1 1 ) ( digits arg1 ) ( reduce #3 0 + ) ( self #4 ) ( + 1
#5 ) ( if #2 0 #6 ) ( lambda1 #7 ) ( sort a ) ( len a ) ( / #10 2 ) ( deref #9 #11 ) ( invoke1 #8 #12 c ) ( -
#13 ) ( - b d ) ( - b d ) ( * #15 #16 ) ( * #14 #17 ) ( + #18 ) ( sqrt #19 ) ( floor #20 )

**Question**: Given numbers a , b , c and e , let d be c , reverse digits in d , let a and the number
in the range from 1 to b inclusive that has the maximum value when its digits are reversed be the
coordinates of one end and d and e be the coordinates of another end of segment f , find the length
of segment f squared
**TP-N2F(correct)**:
( digits c ) ( reverse #0 ) ( * arg1 10 ) ( + #2 arg2 ) ( lambda2 #3 ) ( reduce #1 0 #4 ) ( - a #5 ) ( digits
c ) ( reverse #7 ) ( * arg1 10 ) ( + #9 arg2 ) ( lambda2 #10 ) ( reduce #8 0 #11 ) ( - a #12 ) ( * #6 #13
) ( + b 1 ) ( range 0 #15 ) ( digits arg1 ) ( reverse #17 ) ( * arg1 10 ) ( + #19 arg2 ) ( lambda2 #20
) ( reduce #18 0 #21 ) ( digits arg2 ) ( reverse #23 ) ( * arg1 10 ) ( + #25 arg2 ) ( lambda2 #26 ) (
reduce #24 0 #27 ) ( ¿ #22 #28 ) ( if #29 arg1 arg2 ) ( lambda2 #30 ) ( reduce #16 0 #31 ) ( - #32 e
) ( + b 1 ) ( range 0 #34 ) ( digits arg1 ) ( reverse #36 ) ( * arg1 10 ) ( + #38 arg2 ) ( lambda2 #39
) ( reduce #37 0 #40 ) ( digits arg2 ) ( reverse #42 ) ( * arg1 10 ) ( + #44 arg2 ) ( lambda2 #45 ) (
reduce #43 0 #46 ) ( ¿ #41 #47 ) ( if #48 arg1 arg2 ) ( lambda2 #49 ) ( reduce #35 0 #50 ) ( - #51 e )
( * #33 #52 ) ( + #14 #53 )
**LSTM(wrong)**:
( - a d ) ( - a d ) ( * #0 #1 ) ( digits c ) ( reverse #3 ) ( * arg1 10 ) ( + #5 arg2 ) ( lambda2 #6 ) (
reduce #4 0 #7 ) ( - #8 e ) ( + b 1 ) ( range 0 #10 ) ( digits arg1 ) ( reverse #12 ) ( * arg1 10 ) ( + #14
arg2 ) ( lambda2 #15 ) ( reduce #13 0 #16 ) ( digits arg2 ) ( reverse #18 ) ( * arg1 10 ) ( + #20 arg2 )
( lambda2 #21 ) ( reduce #19 0 #22 ) ( ¿ #17 #23 ) ( if #24 arg1 arg2 ) ( lambda2 #25 ) ( reduce #11
0 #26 ) ( - #27 e ) ( * #9 #28 ) ( + #2 #29 )

### A.6 UNBINDING RELATION VECTOR CLUSTERING

We run K-means clustering on both datasets with $k = 3, 4, 5, 6$ clusters and the results are displayed in Figure 4 and Figure 5. As described before, unbinding-vectors for operators or functions with similar semantics tend to be closer to each other. For example, in the MathQA dataset, arithmetic operators such as *add, subtract, multiply, divide* are clustered together at middle, and operators related to geometry such as *square* or *volume* are clustered together at bottom left. In AlgoLisp dataset, basic arithmetic functions are clustered at middle, and string processing functions are clustered at right.

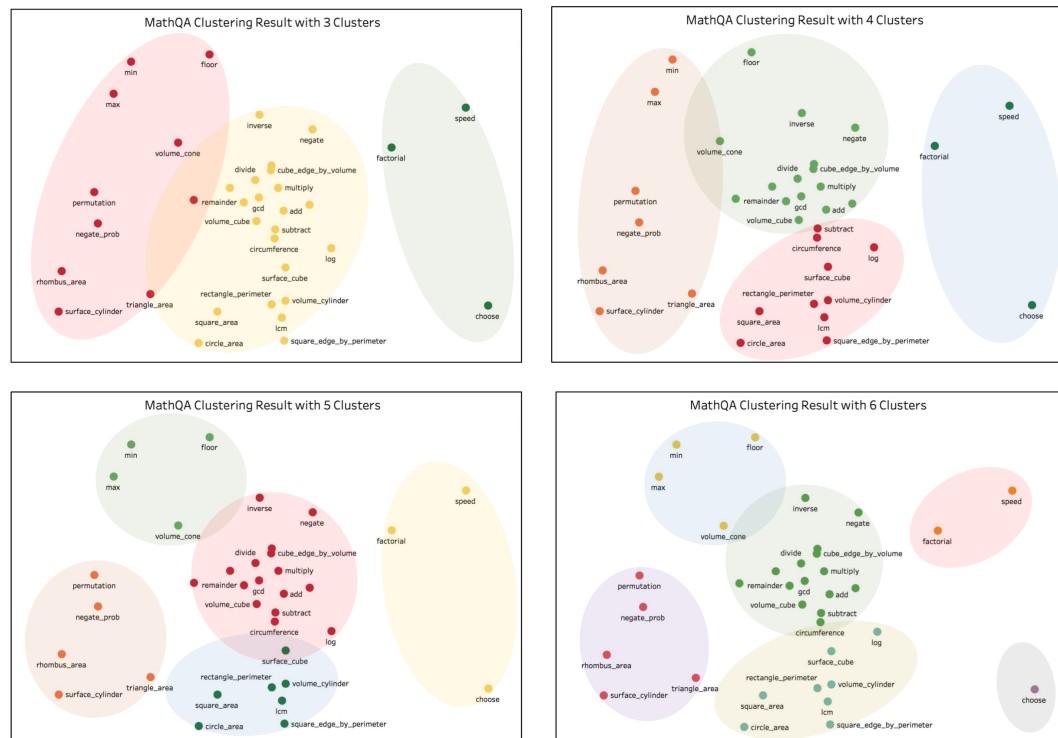

Figure 4: MathQA clustering results

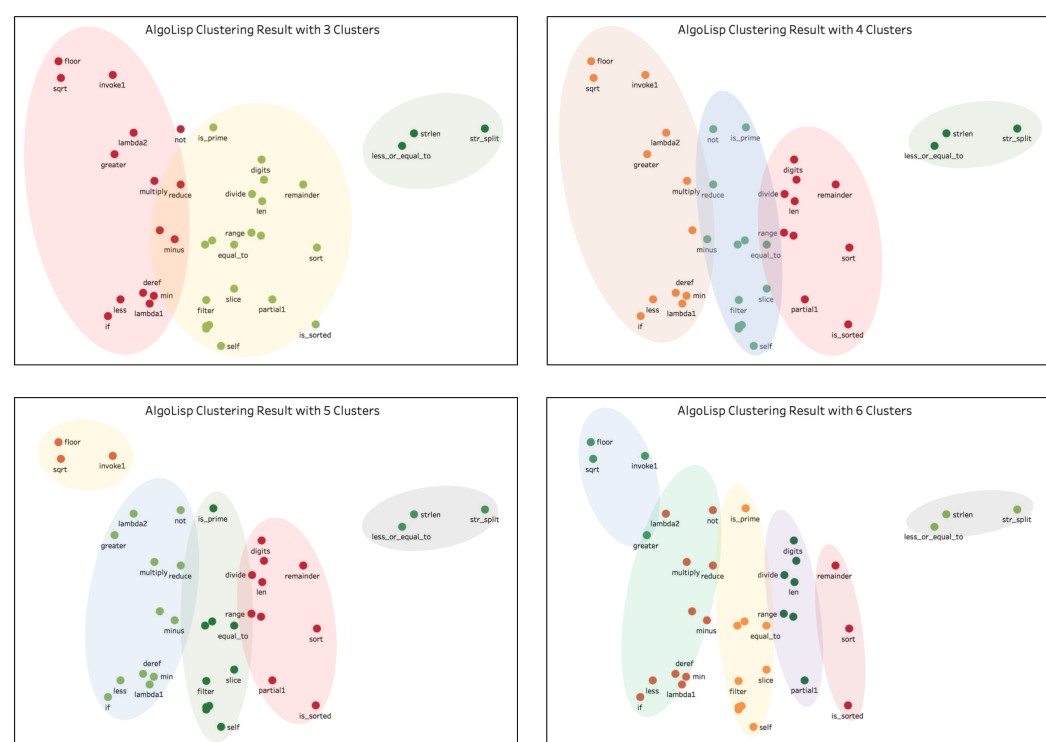

Figure 5: AlgoLisp clustering results

