# OpenReview forum: "Natural- to formal-language generation using Tensor Product Representations"
_ICLR.cc/2020/Conference — Reject_

### Official Review · AnonReviewer2 · 2019-10-24
**Official Blind Review #2**

**Rating:** 8

**Review:**

This paper considers the challenging problem of learning to generate programs from natural language descriptions: the inputs are sequences of words describing a task and the outputs are corresponding programs solving the task. The proposed approach elegantly relies on tensor product representations. Inference with the proposed model is done in 3 steps: (1) encode the symbolic information present in the text data as a TPR, (ii) maps the input TPR to an output TPR encoding the symbolic relations of the output programs (here the authors use a simple MLP), and (iii) decode the output TPR into an actual program. The parameters of the models used in the 3 steps are learned jointly. For step (iii), the authors proposes a novel way of encoding an n-ary relation into a TPR which facilitates the recovery of the relation's arguments using unbinding operations: this is a neat trick (though I think it increases the number of parameters and may limit the expressiveness of the TPR, since reaching "full-rank" of the TPR will occur faster than with the encoding used in [Smolensky et al., 2016]). Experiments on two datasets demonstrate the validity of the approach.

The paper is very well written and easy to follow. The idea seems original and well executed but I think the experimental section could be improved. In particular, adding/reporting stronger baselines to the comparison would straighten the paper. I also feel some relevant literature may be missing from the related work. Nonetheless, I think it is a good paper which will be relevant to the community, I thus recommend acceptance.

* Comments / Questions *

- Section 3.1.1: if I understand correctly, the length of the sequence affects the rank of the TPR. Could that be a problem in practice? E.g., the capacity of the TPR could likely be saturated quickly for long sequences?

- Section 3.2.1: the filler vector f_t = Fu is computed as a convex combination of the learned filler vectors. Is it a design choice to choose a convex combination rather than taking the column corresponding to the argmax of the vector u? Or is it because otherwise the model can not be trained using the classical backprop approach?

- The results of the Seq2Tree+Search model from (Bednarek et al. (2019)) is not reported in Table 2. Why? I believe it should be included (it is ok that it outperforms the proposed method. In addition you can maybe identify clear advantages of your method illustrating a trade-off, e.g., running time, end-to-end, scalability ...).

- A more thorough ablation study could also improve the strength of the experiments. For example, do you know to which extent the attention model in the decoder is necessary to achieve good performances?

- I am not very familiar with the literature but it seems some relevant work may be missing from the review. In particular, I believe there are many papers tackling the problem of learning programs from input output examples or execution traces, e.g. "DeepCoder: Learning to Write Programs", "Neural Turing machines",  "Inferring algorithmic patterns with stack-augmented recurrent nets", "Inferring and Executing Programs for Visual Reasoning", "Learning to infer graphics programs from hand-drawn images"... This list is by no means meant to be exhaustive in any way, just to illustrate a large body of work that seems relevant to the present paper (even though I understand that those papers do not consider natural language description as inputs).

* Typos *

- Eq. (5) Should be r' instead of r_i' (?)


**Experience Assessment:**

I do not know much about this area.

**Review Assessment: Checking Correctness Of Derivations And Theory:**

I assessed the sensibility of the derivations and theory.

**Review Assessment: Checking Correctness Of Experiments:**

I assessed the sensibility of the experiments.

**Review Assessment: Thoroughness In Paper Reading:**

N/A

---

> ### Author Response · Authors · 2019-11-13
> **Response to Reviewer #2**
>
> Thank you very much for your strong recommendation for accepting the paper. We share your excitement about the novelty and impact of the proposed methods, and we would like to provide additional technical details below to address some of your questions.
>
> 1. "length of the sequence":
> The length of the input or output sequence does not affect the order of the corresponding TPR. In the decoder, the order-3 TPR is always of the same "size" (rank): it represents just a single relational tuple, since these are generated one at a time (Sec. 3.1.2). In the encoder, the order-2 TPR of a NL word sequence is the sum of each word's single TPR (Sec. 3.1.1). For problems consisting of a longer sequence of words, the TPR produced by the encoder is intuitively more 'densely-packed' (literally, higher rank, as you say), but it can apparently still adequately represent all the information in the problem needed to correctly generate even quite lengthy output sequences.
>
> For example, the following is generated correctly by our model (55 tuples) but wrong by the baseline (LSTM).
>
> Question: given numbers a , b , c and e , let d be c , reverse digits in d , let a and the number in the range from 1 to b inclusive that has the maximum value when its digits are reversed be the coordinates of one end and d and e be the coordinates of another end of segment f , find the length of segment f squared.
>
> TP-N2F generated tuples (correct):
>
> ( digits c ) ( reverse 0 ) ( * arg1 10 ) ( + 2 arg2 ) ( lambda2 3 ) ( reduce 1 0 4 ) ( - a 5 ) ( digits c ) ( reverse 7 ) ( * arg1 10 ) ( + 9 arg2 ) ( lambda2 10 ) ( reduce 8 0 11 ) ( - a 12 ) ( * 6 13 ) ( + b 1 ) ( range 0 15 ) ( digits arg1 ) ( reverse 17 ) ( * arg1 10 ) ( + 19 arg2 ) ( lambda2 20 ) ( reduce 18 0 21 ) ( digits arg2 ) ( reverse 23 ) ( * arg1 10 ) ( + 25 arg2 ) ( lambda2 26 ) ( reduce 24 0 27 ) ( > 22 28 ) ( if 29 arg1 arg2 ) ( lambda2 30 ) ( reduce 16 0 31 ) ( - 32 e ) ( + b 1 ) ( range 0 34 ) ( digits arg1 ) ( reverse 36 ) ( * arg1 10 ) ( + 38 arg2 ) ( lambda2 39 ) ( reduce 37 0 40 ) ( digits arg2 ) ( reverse 42 ) ( * arg1 10 ) ( + 44 arg2 ) ( lambda2 45 ) ( reduce 43 0 46 ) ( > 41 47 ) ( if 48 arg1 arg2 ) ( lambda2 49 ) ( reduce 35 0 50 ) ( - 51 e ) ( * 33 52 ) ( + 14 53 )
>
> LSTM generated Lisp-code (incorrect):
>
> ( - a d ) ( - a d ) ( * 0 1 ) ( digits c ) ( reverse 3 ) ( * arg1 10 ) ( + 5 arg2 ) ( lambda2 6 ) ( reduce 4 0 7 ) ( - 8 e ) ( + b 1 ) ( range 0 10 ) ( digits arg1 ) ( reverse 12 ) ( * arg1 10 ) ( + 14 arg2 ) ( lambda2 15 ) ( reduce 13 0 16 ) ( digits arg2 ) ( reverse 18 ) ( * arg1 10 ) ( + 20 arg2 ) ( lambda2 21 ) ( reduce 19 0 22 ) ( > 17 23 ) ( if 24 arg1 arg2 ) ( lambda2 25 ) ( reduce 11 0 26 ) ( - 27 e ) ( * 9 28 ) ( + 2 29 )
>
> 2. "convex combination":
> Yes, the main reason for using a weighted combinations of fillers and roles in the encoder is that argmax is not differentiable. Additionally, in other work, we have seen that networks can effectively use the blending of role vectors, performing less well when the blend is replaced by the single argmax role vector.
>
> 3. "Seq2Tree+search results":
> Thanks for the suggestion. The Seq2Tree + Search model is proposed in the original dataset paper (Polosukhin and Skidanov, 2018). For fair comparison, Table 2 shows results without beam search across the board. The accuracy for Seq2Tree without beam search is $61\%$ on the full testing dataset, while ours is $84.02\%$. We did not implement beam search in all the models compared in Table 2, and we have only the authors' reported value for Seq2Tree + Search, which is $85.8\%$. We do not know the potential benefit of beam search for TP-N2F because the output is not a sequence of tokens, but a sequence of full relational tuples, and it is not yet clear how to implement beam search effectively at that level. This is interesting future work.
>
> 4. "ablation study":
> Thanks for the suggestions on this. We actually did many different ablation studies. We tried to use LSTM to decode an entire relational tuple one at a time, i.e., use three different MLPs on each hidden state of the LSTM to predict one relation and two arguments. However, the accuracy is about $15\%$ lower than TP-N2F on the MathQA dataset. We assume that the TPR structure is important for decoding relational tuples, especially relational tuples for reasoning which contain rich structural information.
>
> 5. "references":
> Thank you for the suggestion of discussing the relation between our approach to program synthesis and others. A key difference is the complete absence of any use of symbolic computation in our approach. This makes our approach and others less readily comparable. Were space limits not so severe, we would have liked to follow your suggestion and attempted comparisons.
>
> 6. "typo":
> Yes, thank you: you correctly spotted a typo in (5), which we ourselves only caught after the paper was submitted.

---

### Official Review · AnonReviewer3 · 2019-10-29
**Official Blind Review #3**

**Rating:** 3

**Review:**

This paper proposes a sequence-to-sequence model for mapping word sequences to relation-argument-tuple sequences. The intermediate representation (output of the encoder) is a fixed-dimensional vector. Both encoder and decoder internally use a tensor product representation. The experimental results suggest that the tensor product representation is helpful for both the encoder and the decoder. The paper is interesting and the experimental results are positive, but in my opinion the exposition could use some substantial work. Fixing the most substantial flaws in the exposition would be sufficient to warrant an accept in my view.


Major comments:

I found the mix of levels of detail in the model specification in section 3 confusing. It would be extremely helpful to have a straightforward high-level mathematical description of the key parts of the encoder, mapping (which could be considered part of the encoder), and decoder in standard matrix-vector notation. While equations (7), (8), (9), (10), (11) and appendix A.2 go some way toward this, key high-level details seem to be missing, and I feel like the exposition would benefit from simply stating the matrix-vector operations that are performed in addition to describing their interpretation in terms of the semantics of the tensor product representation. Specific examples are noted below.

It would be helpful to be explicit about the very highest-level structure of the proposed model. If I understand correctly, it is a probabilistic sequence-to-sequence model mapping a word sequence to a probability distribution over relation-argument-tuple sequences. It uses an encoder-decoder architecture with a fixed-dimensional intermediate representation, and an autoregressive decoder using attention. Both the encoder and decoder are based on the tensor product representation described in section 2. Stating these simple facts explicitly would be extremely helpful.

Especially for the encoder, the learned representation is so general that there seems to be no guarantee that the learned roles and fillers are in any way related to the syntactical / semantic structure that motivates it in section 2. There doesn't seem to be any experimental investigation of the learned TPR in the encoder. If I understand correctly, the way encoder roles and fillers are computed and used is symmetric, meaning that the roles and fillers could be swapped while leaving the overall mapping from word sequences to relation-argument-tuple sequences unchanged. This suggests it is not possible to interpret the role and filler vectors in the encoder in an intuitive way.


Minor comments:

In section 2, "R is invertible" should strictly be "R has a left inverse".

In section 3.1.1, the claim that "we can hypothesize to approximately encode the grammatical role of the token and its lexical semantics" is pretty tenuous, especially given the apparent symmetry between learned roles and fillers in the encoder and given the lack of experimental investigation of the meaning of the learned encoder roles and fillers.

In section 3.1.2, my understanding is that the relation-argument tuple (R, A_1, A_2, A_3), say, is treated as a sequence of 3-tuples: (A_1, R, 1), (A_2, R, 2), (A_3, R, 3). Each of these 3-tuples is then embedded using learned embeddings (separate embeddings for argument, relation and position). If correct, it would be helpful to state this explicitly.

In section 3.1.2, it is stated that contracting a rank-3 tensor with a vector is equivalent to matrix-vector product, which is not the case.

In section 3.1.3, both high-level and low-level details of the MLP module are omitted. High-level, I presume that the matrix output by the encoder is reshaped to a large vector, the MLP is applied to this vector to produce another vector, then this is reshaped to a rank-3 tensor to input to the decoder. It would be helpful to state this. Low-level, the number of layers, depth and activation function of the MLP should be specified somewhere, at least in the appendix.

Did the authors consider using a bidirectional LSTM for the encoder? This might improve performance.

In section 3.1.2 and appendix A.2, why use the LSTM hidden state for subsequent processing rather than the LSTM output (which would be more conventional). The LSTM output is defined in appendix A.2 but appears not to be used for anything. Please clarify in the paper.

Did the authors consider passing the output of the reasoning MLP into every step of the tuple LSTM instead of just using it to initialize the hidden state?

It would be helpful to state the rank of the tensors H, B, etc in section 3.2.2.

In section 3.2.2, what does "are predicted by classifiers over the vectors..." mean? This seems quite imprecise. What is the form of the classifier? My best guess is that the vector a_i^t is passed through a small MLP with a final softmax layer which outputs a probability distribution over the 1-of-K representation of the argument. The main text says "more details are given in the appendix", but appendix A.2 just has "Classifier(a_1^t)". Please clarify in the paper.

What is the attention over in equation (9)? Attention needs at least two arguments, the query and the sequence being attended to. It seems that (9) only specifies one of these. It would also be helpful to be explicit about the form of attention used.

What is f_linear in (11)?

It seems unnecessarily confusing to switch notation for the arguments from A_1 in section 3.1.2 to a r g_1 in section 3.2.2, and similarly for the relations.

For the decoder tuple LSTM, how exactly is the previous relation-argument tuple (R, A_1, A_2, A_3), say, summarized? Are each of R, A_1, A_2 mapped to a vector, these vectors concatenated, then passed into the LSTM? Or is the positional decomposition into (A_1, R, 1), ... used? Please clarify in the paper.

Based on section 3.3, it seems that the model assumes that, in the decomposition of (R, A_1, A_2, A_3) into a sequence (A_1, R, 1), (A_2, R, 2), (A_3, R, 3) of 3-tuples at each decoder output step, the three 3-tuples are conditionally independent of each other and the three entries of each 3-tuple are conditionally independent of each other. Is this indeed assumed? If so, it would be helpful to state this explicitly. It seems like this is likely not true in practice.

Section 3.3 refers to "predicted tokens". Where are these predicted tokens in (9), (10) or (11)?

In section 3.3, it seems the loss at each decoder step is the log probability of the relation-argument tuple at that step. Thus, by the autoregressive property, the overall loss is the log probability of the sequence of relation-argument tuples. If so, it would be helpful to state both these facts explicitly.

Section 3 seems to be missing a section, which is how decoding is performed at inference time. For the output of the decoder at each step, is random sampling used, if so with a temperature, or is greedy decoding (selecting the most likely class, equivalent to a temperature of 0) used? Also, what is done if decoding outputs different R's for (A_1, R, 1), (A_2, R, 2), (A_3, R, 3)? The three R values here should be equal in order for this to represent a relation-argument tuple (R, A_1, A_2, A_3), but there is no guarantee the model will respect this constraint.

Unless I missed it (apologies if so), many experimental architecture details were omitted. For example, how many hidden cells were used for the LSTMs, etc, etc? These should at least be stated in the appendix.

It would be interesting to investigate how long input / output sequences need to be before the fixed-dimensional internal representation breaks down.

In section 4.1.1, it was not clear to me what "noisy examples" means. Does this mean that the dataset itself is flawed, meaning that the reference sequence of operations does not yield the reference answer? Please clarify in the paper.

In table 1, please state the total size of the fixed-dimensional intermediate representation for all systems. This seems crucial to ensure the systems can be meaningfully compared.

In figure 4, left figure, the semantic classes don't apper to be very convincingly clustered. (And it seems like K-means clustering could easily have selected a different clustering given a different random initialization.)

In appendix A.2, mathematical symbols are essentially meaningless without describing what they mean in words. Please explain the meaning of all the symbols that are not defined in terms of other symbols, e.g. w^t, T_{t-1}, ..., f_s m (is this softmax???), f_l i n e a r (what does this mean?), C o n t e x t, C l a s s i f i e r, etc, etc. C o n t e x t in particular doesn't even have a hint of a definition.

In (19) and (27), why would a temperature parameter be helpful? This can be absorbed as an overall multiplicative factor in the weight matrix of the previous linear layer. Is this temperature parameter learned during training (I presume so)? Please clarify in the paper.

Usually * is used for convolution, not simple multiplication (e.g. equation (17)).

Throughout the main body and appendix, there are lots of instances of poor spacing. For example, $f_{linear}$ should be written as something like $f_\text{linear}$ in latex to avoid it coming out as l i n e a r (which literally interpreted means l times i times n times e, etc). Please fix throughout.

**Experience Assessment:**

I do not know much about this area.

**Review Assessment: Checking Correctness Of Derivations And Theory:**

I carefully checked the derivations and theory.

**Review Assessment: Checking Correctness Of Experiments:**

I assessed the sensibility of the experiments.

**Review Assessment: Thoroughness In Paper Reading:**

I read the paper thoroughly.

---

> ### Author Response · Authors · 2019-11-13
> **Response to the major comments of Reviewer #3**
>
> Your very comprehensive comments on our work and the excellent advice on the presentation are greatly appreciated. We believe we have addressed all of your comments in the new version of the paper that we have uploaded to OpenReview, which is heavily revised and much improved, thanks to your suggestions and those of the other reviewers. We have also added an important new mathematical Section A.3 to the Appendix, showing that a successfully trained model will have learned to produce, for the decoder, order-3 TPRs that have the form assumed in the network design (Eq. 3, p. 4).
>
> In this comment, we address your two major points; the 27 minor points are taken up in a follow-on comment.
>
> 1. "high-level structural description of the model":
> The revised version includes: "TP-N2F encodes the natural-language symbolic structure of the problem in an input vector space, maps this to a vector in an intermediate space, and uses that vector to produce a sequence of output vectors that are decoded as relational structures. Both input and output structures are modeled as Tensor Product Representations (TPRs) (Smolensky, 1990). During encoding, NL-input symbolic structures are encoded as vector space embeddings using TPR `  binding' (following  "Palangi, et al., AAAI 2018"); during decoding, symbolic constituents are extracted from structure-embedding output vectors using TPR `  unbinding' (following "Huang, et al., NAACL 2018; AAAI 2019")." [p. 1] We have also added a more specific but still high-level description in the opening 2 paragraphs of Sec. 3, p. 3.
>
> 2. "symmetry between roles and fillers in encoder":
> On the important issue you raise here, the revised paper includes footnote 3, p. 3: "Although the TPR formalism treats fillers and roles symmetrically, in use, hyperparameters are selected so that the number of available fillers is greater than that of roles. Thus, on average, each role is assigned to more words, encouraging it to take on a more general function, such as a grammatical role." Our model uses 150 fillers and 50 roles. The text on p. 3 also now states: "the mechanism closely follows that of "Palangi, et al., AAAI 2018", and we hypothesize similar results: the role and filler approximately encode the grammatical role of the token and its lexical semantics, respectively." Exploring inductive biases that explicitly encourage the roles to be used grammatically is a direction of future work.

---

> > ### Author Response · Authors · 2019-11-13
> > **Response to the minor comments of Reviewer #3**
> >
> > We really appreciate your careful suggestions and valuable comments very much! Your 27 minor comments greatly aided us in improving the exposition; we hope that you would find the thoroughly revised paper (uploaded to OpenReview) much clearer. Because of the limitation of space, we apologize for some unclear points and formatting issues. We will address the main questions below.
> >
> > For minor comments 1, 2, 3, 4, 7, 9, 13, 15, 17, 20, 22, 23, 24, 26, 27 we updated the paper based on these suggestions about clarification and formatting. The mathematics has been completely re-set, expanded to completely define all quantities in the model, and augmented with English explanation.
> >
> > 5. "details of the MLP":
> > The details of the MLP are now given in Sec. A.2.2 [p. 12]. Each layer is a linear layer followed by a tanh activation function. We tested the number of layers from 1 to 5. With 1 and 2 layers of the MLP, the performance is roughly the same and with more layers, the performance drops. We use the best result, from one linear layer with tanh.
> >
> > 6. "bidirectional LSTM for the encoder":
> > We tested bidirectional LSTMs, but did not get significant improvement and the best models are trained with unidirectional LSTMs.
> >
> > 8. "using the output of reasoning MLP in tuple-LSTM":
> > Although the reported model passes the output of the reasoning MLP only to the first time-step of the tuple-LSTM, the decoder correctly produces lengthy sequences of tuples: see the last example now given in Sec. A.5 (and in the reply to Review #2), a correct sequence of 55 tuples. However, the performance may well be improved further by making the output of the reasoning MLP available to the decoding LSTM at every time step; thank you for suggesting this variation, which we will test in future work.
> >
> > 10. "classifier":
> > Sorry for the missing details. As now spelled out in Sec. A.2.3, Eqs. 46-51 [p. 13], we use a linear layer followed by a softmax function separately on the relation vector and the 2 argument vectors to compute the probability distribution over all possible symbolic relations and arguments. Generation is done greedily.
> >
> > 11. "attention":
> > The attention version we used is from "Effective Approaches to Attention-based Neural Machine Translation", Luong, et al. (2015), which is now described in detail in Sec. A.2.3 (Eqs. 38-40, p. 13).
> >
> > 12. "$f_{linear}$ in (11)":
> > As now spelled out in Eq. 43 of Sec. A.2.3, the $f_{linear}$ in (11) is a simple linear function to generate the unbinding relation vector. The linear function operates on the sum of $\textbf{B}_1$ and $\textbf{B}_2$ (the tensor product between the relation vector and each argument vector) to produce an unbinding vector of a relation that unbinds both arguments (Eqs. 44-45).
> >
> > 14. "the input of the decoder":
> > The input at each timestep of the decoder Tuple-LSTM is the concatenation of the relation and argument embedding vectors of the tuple generated at the previous timestep.
> >
> > 16. "predicted tokens in (9), (10) or (11)" and 18. "decoding at inference time":
> > The text now states: "the softmax probability distribution over symbolic outputs is computed for relations and arguments separately. In generation, the most probable symbol is selected." [p. 6] and spelled out in Eqs. 49-51 of Sec. A.2.3 [p. 13].
> >
> > 19. "details of the experimental architecture setting":
> > These are described in the results and discussion section, and in more detail in the Appendix A.1. We could add  details of all ablation studies, and we will publish our code on github if the paper is accepted.
> >
> > 21. "the noisy examples":
> > The revised version defines an example as noisy if "the execution script returns the wrong answer when given the ground-truth pro-gram" [p. 7].
> >
> > 25. "a temperature parameter be helpful":
> > As you say, the temperature is a factor scaling the weight matrix. As now stated in Sec. A.2.1, p. 11, in the model, it is fixed (to 0.1 in the experiment). The model trained faster with this factor.

---

### Official Review · AnonReviewer4 · 2019-11-04
**Official Blind Review #4**

**Rating:** 3

**Review:**

The authors propose a binding-unbinding mechanism for translating natural language to formal language. The idea is good and novel. As far as I know, this is indeed the first work for handling this task using binding-unbinding mechanism. The experimental results also look promising in compared with the exsiting models. However, the designed specific neural network does not support the claimed binding-unbinding theory very well. Moerover, there seem to be some errors about the correctness of the theory (See the first point below).

Firstly, in the last paragraph of Section 2, the authors claim that the role matrix $R$ would be invertible such that there exists a matrix $U = R^{-1}$ such that the fillers would be recovered. However, $R$ is defined as a non-square matrix in the previous paragraph. How can a non-square matrix be invertible?

Secondly, the design of the specific neural network cannot describe the theory behind proposed binding-unbinding mechanism properly. The authors try to interpret the design of the neural networks using the concepts in the proposed binding-unbinding theorybut are not convincible.  In Section 2, the basics of binding-unbinding are introduced and many mathematical properties are required to make the binding-unbinding work. However, all the parameters/variables in the neural networks are freely designated and are not correlated to each other, thus they cannot work together to meet the requirements in the binding-unbinding mechanism. According to my understanding, at least there should be some direct connections between the parameters in the encoder and decoder. For example, is there any restriction on the parameters in encoder and decoder respectively to reflect the property $UR=I$ as in Section 2.

Lastly, in the encoder part, the role and filler are learned in an unsupervised without any evidence. The input for the decoder is an "assumed" TPR, thus the only evidence from the objective function are cut-off by the assumed TPR. Given that there are no other connections between encoder and decoder, the design of the encoder cannot learn role and filler properly.

Other suggestions:
The natural language to formal language problem is named semantic parsing in natural language processing field. In semantic parsing problem, langugae to programatic language is a typical task. I would recommend include some references in semantic parsing.


**Experience Assessment:**

I have read many papers in this area.

**Review Assessment: Checking Correctness Of Derivations And Theory:**

I carefully checked the derivations and theory.

**Review Assessment: Checking Correctness Of Experiments:**

I assessed the sensibility of the experiments.

**Review Assessment: Thoroughness In Paper Reading:**

I read the paper at least twice and used my best judgement in assessing the paper.

---

> ### Author Response · Authors · 2019-11-13
> **Response to Reviewer #4**
>
> We would like to thank you for the constructive comments and helpful suggestions. We have done our best to address each of your comments in the heavily revised version of the paper that we have uploaded to OpenReview. Regarding your specific points:
>
> 1."non-square matrix": This has been corrected in the revised paper, which now states "if the embeddings of the roles are linearly independent ... the role matrix $\mathbf{R}$ has a left inverse" [p.2]. The case of non-linear-independence is discussed in the new Appendix Sec. A.3, which states, citing the new reference Anonymous (in prep.), that "even when the number of unbinding vectors exceeds the dimension of their space by a factor of 2 or 3 (which applies to the TP-N2F models presented here), there is a set of role vectors $\{ \mathbf{r}_k \}_{k \in K}$ approximately dual to $\{ \mathbf{r}'_k \}_{k \in K}$, such that $\mathbf{r}_l^\top \mathbf{r}_j' = \delta_{lj} \: \forall l, j \in K$ holds to a good approximation" [p.14]
>
> 2."binding-unbinding mechanism properly": We clarify this potentially confusing issue in the revised paper. Regarding the relation between the role and unbinding vectors for the encoder and decoder: "we will make use of both TPR binding using the tensor product with role vectors $\mathbf{r}_i$ and TPR unbinding using the tensor inner product with unbinding vectors $\mathbf{u}_j$. Binding will be used to produce the order-2 tensor $\mathbf{T}_S$ embedding of the NL problem statement. Unbinding will be used to generate output relational tuples from an order-3 tensor $\mathbf{H}$. Because they pertain to different representations (of different orders in fact), the binding and unbinding vectors we will use are not related to one another." [p.2] (Reviewer 3's comment gives a good description for our model.) The job of the MLP between the encoder and the decoder is to map the order-2 natural-language-structure TPR to the order-3 formal-language-structure (relational-tuple) TPR.
>
> 3."the input to the decoder is an 'assumed' TPR": Regarding the status of the "assumed "TPR form of the input to the decoder, $\mathbf{H}$, the revised paper states:
> "In the model, the order-3 tensor $\mathbf{H}$ of Eq. 3 has a different status than the order-2 tensor $\mathbf{T}_S$ of Sec. 3.1.1. $\mathbf{T}_S$ is a TPR by construction, whereas $\mathbf{H}$ is a TPR as a result of successful learning. To generate the output relational tuples, the decoder assumes each tuple has the form of Eq. 3, and performs the unbinding operations which that structure calls for. In Appendix Sec. A.3, it is shown that, if unbinding each of a set of roles from some unknown tensor $\mathbf{T}$ gives a target set of fillers, then $\mathbf{T}$ must equal the TPR generated by those role/filler pairs, plus some tensor that is irrelevant because unbinding from it produces the zero vector. In other words, if the decoder succeeds in producing filler vectors that correspond to output relational tuples that match the target, then, as far as what the decoder can see, the tensor that it operates on is the TPR of Eq. 3."[p.4]
>
> 4."the encoder cannot learn role and filler properly": The fillers and roles in the encoder are learned through end2end supervised training on natural-language-input/formal-language output pairs, following the successful use of this technique for question-answering in "Palangi, et al. (AAAI 2018)".
>
> 5."include some references in semantic parsing": This is an excellent suggestion, which we have followed in the Related Work Sec. 5 [p. 8], although not to the extent we would have liked due to the length limit. Our work is not literally semantic parsing in the narrow sense, since the output is not the meaning of the input, but rather a reasoning process for solving the problem expressed by the input. But we agree there is an important connection, and semantic parsing would be an excellent application of the model.
>
> Again, thank you for your comments and we welcome further helpful suggestions and questions to improve the paper.

---

> > ### Comment · AnonReviewer4 · 2019-11-15
> > **Concerns about the 3rd point**
> >
> > Thanks for your clarifications. But I still have somes concerns about the 3rd point, i.e. the assumed TPR.
> >
> > Why don't you just remove the reasoning MLP layer? If the MLP layer is removed, your theory (Equations 1-5) is perfect. I agree that the learned TPR could be interpreted as having the form of Equation 3. However, I suppose the MLP layer  harms your theory in this model. The reason is follows (I may use some different symbols as in your paper):
> >
> > Let $H_s$ denote the output of the encoder. Then $H_s$ is constructed as $H_s = a_1^s \otimes r^s \otimes p_1^s + a_2^s \otimes r^s \otimes p_2^s$
> > Let $H_r$ denote the output of the MLP reasoning layer, i.e. assumed TPR. Then $H_r$ is learned as $H_r = a_1^r \otimes r^r \otimes p_1^r + a_2^r \otimes r^r \otimes p_2^r$
> >
> > Let $p'$ and $r'$ denote the parameters in the decoder. Then your theory about the binding-unbinding (Equation 4 and 5) takes effect for the pairs of ($p'$, $p^r$) and ($r'$,$r^r$), rather than ($p'$, $p^s$) and ($r'$,$r^s$). So because of the assumed TPR $H_r$, it becomes meanless how $H_s$ is constructed as your theory does not apply to $r^s$ and $p^s$

---

> > > ### Author Response · Authors · 2019-11-15
> > > **Clarification for the concern**
> > >
> > > Thank you for your continued interest in our model.  Your comment starts by  saying that the output of the encoder ($H_s$ in your notation) has the form $\sum_i a_i \otimes r_i \otimes p_i$ , an order-3 tensor. But actually the output of the encoder is an order-2 tensor with the form $\sum_k f_k \otimes r_k$ (Sec. 3.1.1).  In order to produce the order-3 tensor that you describe (Sec. 3.1.2), the MLP is necessary: it converts the order-2 tensor coming out of the encoder into the order-3 tensor that goes into the decoder (Sec. 3.1.3). We called it the "reasoning MLP" because the MLP is supposed to "reason" about the natural-language question and map it to the formal-language program to solve the question.
> > >
> > > If the encoder produced the kind of order-3 tensor you call $H_s$, you are correct that we would not need an MLP between the encoder and the decoder. However the encoder’s job is not to produce the (order-3) encodings of the output relational triples; its job is to produce an encoding of the NL problem statement, which we encode as an order-2 tensor; this was shown to be an effective NL input encoding for QA in Palangi et al. (2018). So the job of the MLP – roughly – is to convert an encoding of the problem (order-2 tensor) into an encoding of the solution (order-3 tensor). It is indispensable for the model. Through this NLP in our model, the natural-language (order-2 tensor) is converted to formal-language (order-3 tensor) in our paper.

---

> > > > ### Comment · AnonReviewer4 · 2019-11-15
> > > > **question about the encoder**
> > > >
> > > > Thanks. Then I have a better understanding about your model. And I also notice you have modified your paper to make it more clear. I have another question. Are $r$ in the encoder and $r'$ in the decoder dual of each other? According to my understanding from your theory, they should be dual of each other. But it seems $r'$ is dual to some implicit vectors encoded in the assumed $H$. If so, the construction of $H_s$ using two LSTM becomes meanlingless. For example, perhaps you may use one LSTMs, product of three LSTMs or product of four LSTMs but get a better result, which, though, are not interpretable by the tensor product representation.

---

> > > > > ### Author Response · Authors · 2019-11-15
> > > > > **Reply to the questions**
> > > > >
> > > > > Thank you for your helpful suggestions, and we hope you find our modifications sufficient for the revised version of this paper to make it more clear.
> > > > >
> > > > > Regarding to the careful questions, the $r'$ in decoder is not the dual of $r$ in the encoder. In encoder, $r$ is the ${\bf role}$ vector of each word in natural-language, which represents more general information such as structural information of the word. In decoder, the $r'$ is the dual vector of ${\bf relation}$ vector in the tuple TPR for formal language (For order-3 TPR $a_i \otimes r_i \otimes p_i$ ,  $a_i$ represents the arguments of a tuple, $r_i$ for relation of a tuple, and $p_i$ for positions of each argument in the tuple).
> > > > >
> > > > > The encoder uses "binding" operation on natural-language TPR (order-2 Tensor); the decoder uses "unbinding" operation on formal-language TPR (order-3 Tensor). During training part, encoder learns to select ${\bf role}$ vectors and ${\bf filler}$ vectors for encoding natural-language via binding; and the decoder learns to use the correct unbinding vectors to unbind the formal-language tuples via unbinding. At each time step $t$, as the decoder needs to decode a order-3 tensor, to decode a binary ${\bf relational}$ tuple, the unbinding module decodes it using the $two$ $steps$ of TPR unbinding ( details are described in Sec. 3.2.2 ). Although the order-3 TPR is called "assumed", the order-3 TPR exists when decoder successes in producing the targeted formal-language tuples (we clarified this at the 3rd point in our original response). We also added a section (Appendix A.3)  in the modified paper for the detailed mathematical explanation about this.
> > > > >
> > > > > During encoding, based on natural-language model of TPR, natural-language is modeled as the order-2 tensor. Each word is represented as the tensor product representation (2-order tensor) of a ${\bf role}$ vector, which represents more general information such as structural information;  and another ${\bf filler}$ vector, which represents more specific information such as semantic information. We use one LSTM for the ${\bf role}$ vector and another LSTM for the  ${\bf filler}$ vector (two LSTM). Based on the theory, two vectors are sufficient to represent the natural-language structure. Furthermore, this order-2 TPR has been shown to be an effective natural-language input encoding for QA in Palangi et al. (2018). In the future, higher order tensors like you mentioned (three LSTM or four LSTM) could be explored to represent different models of natural-language.
> > > > >
> > > > > In this paper, one of our contributions is the learning scheme for learning structure conversion between different TPR (natural-language order-2 tensor to formal-language order-3 tensor). This learning scheme can also be used for either for same structure conversion or different structure conversion. Conversion between same structures can use the first point you mentioned (the corresponding pairs of dual vectors for both binding and unbinding).
> > > > >
> > > > > Thanks again for your interests and questions, and we are happy to answer any further questions you might have.

---

### Decision · Program_Chairs · 2019-12-19

**Decision:**

Reject

**Comment:**

The paper proposed a new seq2seq method to implement natural language to formal language translation.  Fixed length Tensor Product Representations are used as the intermediate representation between encoder and decoder.  Experiments are conducted on MathQA and AlgoList datasets and show the effectiveness of the methods.  Intensive discussions happened between the authors and reviewers.  Despite of the various concerns raised by the reviewers, a main problem pointed by both reviewer#3 and reviewer#4 is that there is a gap between the  theory and the implementation in this paper.  The other reviewer (#2) likes the paper but is less confident and tend to agree with the other two reviewers.